# Structural and functional analysis of vaccinia viral fusion complex component protein A28 through NMR and molecular dynamic simulations

**Chi-Fei Kao**[1☯], **Min-Hsin Tsai**[2☯], **Kathleen Joyce Carillo**[2☯], **Der-Lii Tzou**[2]*, **Wen Chang** [1]*

**1** Institute of Molecular Biology, Academia Sinica, Taipei, Taiwan, **2** Institute of Chemistry, Academia Sinica, Taipei, Taiwan

☯ These authors contributed equally to this work.
* tzougate@gate.sinica.edu.tw (D-LT); mbwen@ccvax.sinica.edu.tw (WC)

**Data Availability Statement:** All relevant data are within the manuscript and its Supporting Information files.

## Abstract

Host cell entry of vaccinia virus (a poxvirus) proceeds through multiple steps that involve many viral proteins to mediate cell infection. Upon binding to cells, vaccinia virus membrane fuses with host membranes via a viral entry fusion protein complex comprising 11 proteins: A16, A21, A28, F9, G3, G9, H2, J5, L1, L5 and O3. Despite vaccinia virus having two infectious forms, mature and enveloped, that have different membrane layers, both forms require an identical viral entry fusion complex for membrane fusion. Components of the poxvirus entry fusion complex that have been structurally assessed to date share no known homology with all other type I, II and III viral fusion proteins, and the large number of fusion protein components renders it a unique system to investigate poxvirus-mediated membrane fusion. Here, we determined the NMR structure of a truncated version of vaccinia A28 protein. We also expressed a soluble H2 protein and showed that A28 interacts with H2 protein at a 1:1 ratio *in vitro*. Furthermore, we performed extensive *in vitro* alanine mutagenesis to identify A28 protein residues that are critical for H2 binding, entry fusion complex formation, and virus-mediated membrane fusion. Finally, we used molecular dynamic simulations to model full-length A28-H2 subcomplex in membranes. In summary, we characterized vaccinia virus A28 protein and determined residues important in its interaction with H2 protein and membrane components. We also provide a structural model of the A28-H2 protein interaction to illustrate how it forms a 1:1 subcomplex on a modeled membrane.

## Author summary

Poxvirus is a large DNA virus family with a complex virus structure and vaccinia virus has been used as a model to study poxvirus biology. It is known that vaccinia virus uses multiple viral attachment proteins to bind to a wide variety of host cells. However, it is still unknown how vaccinia virus executes post-binding membrane fusion with host membranes. Unlike viral fusion proteins identified in other DNA and RNA viruses, the

**Funding:** The work is supported by grants provided by Academia Sinica (AS-IDR-112-02 to W.C.) and National Sciences and Technology Council (110-2320-B-001 -015 -MY3 to W.C.) and (NSTC 111-2113-M-001-014 to D.L.T.). The work is also supported by Academia Sinica Core Facility supported by the Innovative Instrument Projects (AS-CFII-111-214 and AS-NBRPCF-111-201 to D. L.T.) URLs to sponsors' websites: https://www. nstc.gov.tw https://www.sinica.edu.tw The funders had no role in study design, data collection and analysis, decision to publish, or preparation of the manuscript.

**Competing interests:** The authors have declared that no competing interests exist.

components of vaccinia fusion protein complex do not possess any known homology, making it a unique system to explore poxvirus-mediated membrane fusion. Here, we determined the NMR structure of vaccinia A28 protein and demonstrated that A28 interacts with H2 protein at a 1:1 ratio *in vitro*. Furthermore, we identified A28 protein residues that are critical for H2 binding, fusion complex formation, and virus-mediated membrane fusion. Utilizing a combination of biological data and molecular dynamic simulations, we developed a comprehensive computer model showcasing the complete A28-H2 subcomplex structure within viral membranes. This study offers valuable insights for future investigations concerning poxvirus-mediated membrane fusion.

## Introduction

Vaccinia virus (VACV), of Family Poxviridae, has a double-stranded DNA genome of ~190 kilobases [1] that encodes more than 200 proteins [2]. VACV has been deployed to eradicate smallpox and is widely used as a model system for studying other poxviruses. Given the recent outbreak of mpox worldwide, it remains critical to understand VACV biology in order to understand how poxviruses mutate and evolve in hosts to enable future development of preventive and therapeutic measures against pathogenic poxviruses.

VACV replicates in the cell cytoplasm, with virus assembly generating mature virus (MV) particles encompassed by a single membrane derived from endoplasmic reticulum (ER). Additional wrapping of Golgi membrane during virus egress generates an extracellular virus (EV) enclosed by two membranes. MV and EV infect cells through different entry pathways, depending on virus strains and host cell types [3–5]. Despite MV and EV containing different proteins for cell attachment, post-binding fusion between viral and cellular membranes requires an identical set of 11 viral proteins that form a conserved viral entry fusion complex (EFC) [6,7]. The viral EFC contains A16 [8], A21 [9], A28 [10,11], F9 [12], G3 [13], G9 [14], H2 [15], J5 [16], L1 [17], L5 [18] and O3 [19], with each component being critical for VACV particle infectivity, EFC complex formation, and membrane fusion activity. Previous genetic and biochemical experiments have revealed that some components form subcomplexes, such as A28 and H2 [6], G3 and L5 [20], and A16 and G9 [21,22]. Moreover, a recent investigation of EFC component interactions indicated even more complex interactions among EFC components and that the O3 protein interacts with all other EFC components [23]. Super-resolution microscopy of single VACV particles has revealed that the viral EFC clusters at the tips of viral particles, separately from the viral attachment proteins H3, D8 and A27 [24]. Furthermore, a fluorescence dequenching analysis of VACV mutants in which expression of individual EFC components was suppressed indicated that the 11 EFC proteins may act at different stages of the membrane fusion process [25]. Crystal structures of the VACV F9 [12] and L1 [26] proteins have been reported previously. Recently, the crystal structures of G3-L5 [27] and A16-G9 [28] heterodimers were also reported. However, none of these structurally-assessed vaccinia EFC components share homology with any known class I, II or III viral fusion protein [29–31]. Consequently, it remains unclear how the VACV EFC mediates membrane fusion.

Among the VACV EFC components, A28 was first shown to be essential for VACV virus infectivity and EFC formation [6,10,32]. To investigate its structure and functional characteristics, we determined a 3D high-resolution NMR structure of a truncated form of vaccinia A28 protein. Furthermore, we performed a series of biochemical and mutational analyses in order to determine the structural features that are critical for its function in membrane fusion. Given the high sequence homology among poxviral A28 orthologs, our findings on A28 and its

binding partner H2 can be directly applied to related pathogenic orthopox viruses, including mpox and smallpox.

## Results

### Nuclear magnetic resonance (NMR) structure of truncated A28 in solution

We conducted structural and functional analyses of VACV A28 by means of solution NMR spectroscopy. We expressed a truncated form of the A28 protein, tA28 (comprising residues 38–146), in which the N-terminal 37 residues had been deleted (Fig 1A). Purified tA28 was soluble (Fig 1B), but repeated crystallization screenings failed to generate a protein crystal. Instead, we applied Nuclear Overhauser Enhancement (NOE) to determine the structure of tA28 protein in solution at pH6.5. As summarized in Table 1, based on the 968 NOE restraints and 134 dihedral angle restraints, we could derive the most probable molecular structure of tA28. In our structural refinements, we applied an AMBER force field for energy minimization, which introduced solvent molecules such as water and ions into the surroundings of tA28, so that the overall molecular structure of tA28 represents that in the solution phase. Through sequential resonance assignments of tA28 [33], we identified helical structures, β-strands, and interconnecting loop regions in the truncated protein. The overall structural quality of tA28 was significantly improved upon applying AMBER force field refinement. The structural improvements of tA28 in terms of the initial rmsd over the secondary structure are achieved. Overall, tA28 comprises three typical α-helices-α1 (residues 68–71), α2 (109–117), and α3 (136–145) as well as a short $3_{10}$-helix (119–121), and five anti-parallel β-strands-β1 (74–78), β2 (82–87), β3 (90–96), β4 (100–103) and β5 (124–126)—with the remaining residues forming connecting loops (Fig 1C). In addition, we identified two intramolecular disulfide bonds, bridging C75-C112 and C129-C139, (Fig 1D and 1F) based on the $^{13}C^{\beta}$ chemical shifts of the Cystine residues [34,35], consistent with a previous report [32]. For the final ensemble of the 10 lowest-energy structures (Fig 1C), we achieved a high-resolution molecular structure by solution NMR with an average rmsd of 0.9 Å (Table 1).

The tA28 protein structure determined by NMR reveals an α-β-α sandwich structural motif, in which the β-strands are sandwiched by two α-helices (Fig 1E&1F). Multiple hydrogen bonds among the β-strands contribute to stabilizing the tA28 structure, with the two intramolecular disulfide bonds also contributing to protein stability.

### A28 and H2 protein interactions *in vitro*

Next, we analyzed the interactions of A28 with H2 *in vitro*, another component of the VACV EFC, since previous studies had revealed that they form a subcomplex in infected cells [6,15]. To do so, we deployed 2D $^1H$-$^{15}N$ heteronuclear single quantum coherence (HSQC) NMR spectroscopy. First, we expressed a further truncated $^{15}N$-isotope-labeled form of A28 protein, sA28 (residues 56–146), that displays increased solubility and stability relative to tA28 (Fig 1A&1B). All chemical shift patterns in NMR for sA28 (S1 Fig) were consistent with previous assignments for tA28 (residues 38–146) [33], implying that the overall protein conformation remained intact. We also expressed and purified a soluble form of vaccinia H2 protein, sH2 (residues 91–189) (Fig 2A), for the *in vitro* HSQC binding analysis described below.

2D $^1H$-$^{15}N$ HSQC spectroscopy represents a reliable and robust approach for establishing protein-protein interactions with atomic resolution. By labeling one of two binding partners with a $^{15}N$ isotope, specific molecular interactions can be probed based on chemical shift perturbations due to the presence of the binding partner. First, we acquired 2D HSQC spectra of $^{15}N$-isotope labeled sA28 protein at pH 6.5 and deduced the chemical shift assignments in the presence or absence of sH2 at pH6.5 (Fig 2B). We then superimposed the two spectra and

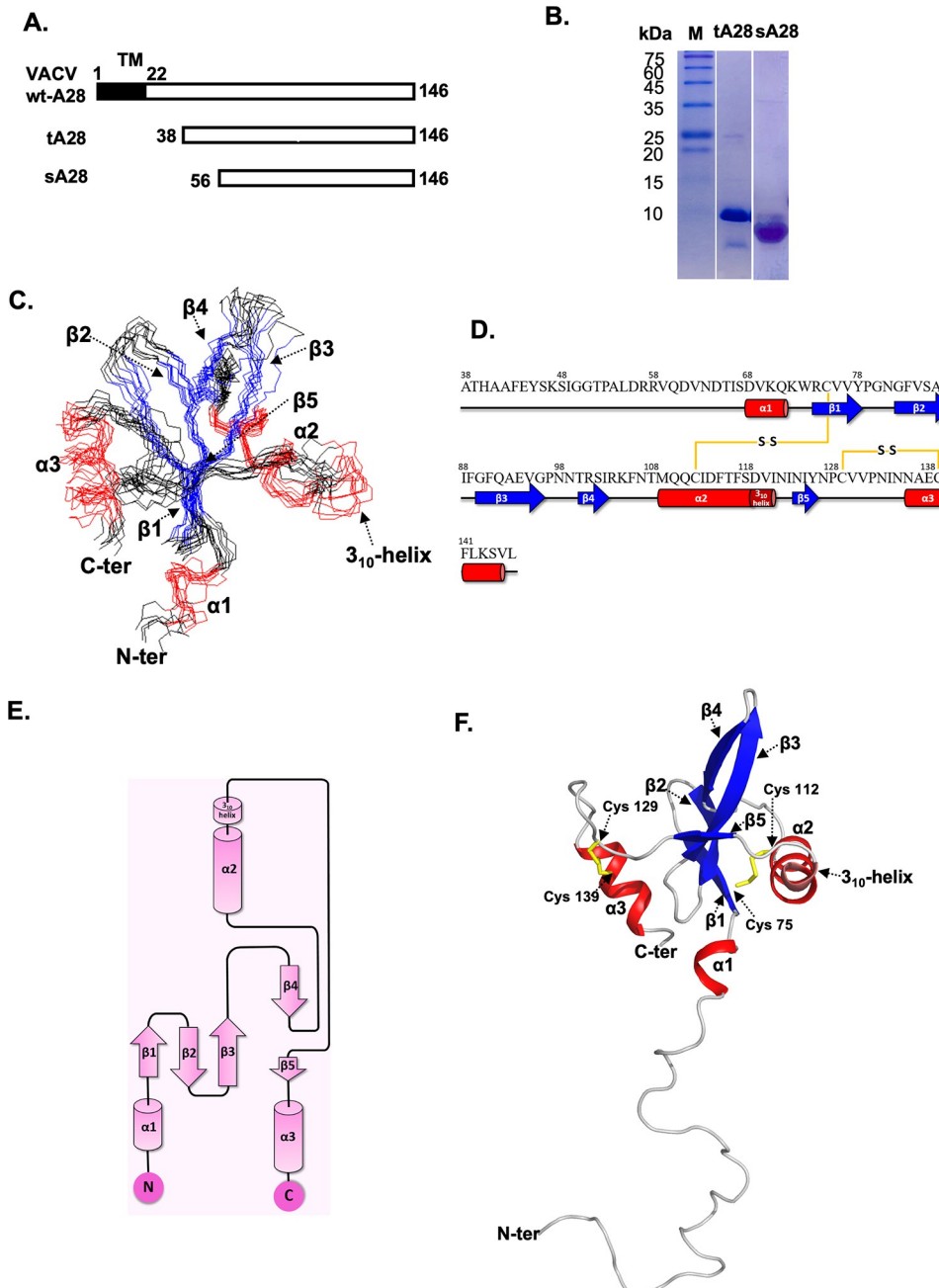

**Fig 1. Solution NMR structural determination of tA28(38–146)** (**A**) Wild-type full-length A28 protein (amino acids 1–146) contains a transmembrane domain (TM) (aa 1–22) at the N-terminus. A truncated A28 (aa 38–146; tA28) and a smaller soluble form of A28 (aa 56–146; sA28) were overexpressed in *Escherichia coli* and purified for this work. (**B**) Coomassie blue staining of an SDS-PAGE gel of purified recombinant tA28 and sA28 proteins. M: protein markers. (**C**) An ensemble of the 10 lowest-energy NMR structures of tA28. (**D**) The secondary structure of tA28 consists of three α-helixes: α1 (aa 68–71), α2 (aa 109–117), α3 (aa 136–145); five β-strands: β1 (aa 74–78), β2 (aa 82–87), β3 (aa 90–96), β4 (aa 100–103) and β5 (aa 124–126); and one $3_{10}$-helix (aa 119–121). The α-helixes, β-strands and random coil are labeled by red cylinders, blue arrows and a dark grey line, respectively. (**E**) tA28 protein topological diagram generated from PDBsum. (**F**) The lowest-energy NMR structure of tA28 at pH 6.5, as determined by NMR spectroscopy. Ribbon representation of the average NMR structure of tA28, with the β-strands (blue) shown as sandwiched by the α2 and α3 α-helixes (red). Two pairs of disulfide bonds, Cys75-Cys112 and Cys129-Cys139, are shown in yellow.

**Table 1. NMR and refinement statistics for vaccinia virus fusion protein tA28.**

|  | tA28 (38–146) |
|---|---|
| **NMR distance and dihedral constraints** |  |
| Distance constraints |  |
| Total NOE | 968 |
| Intra-residue | 344 |
| Inter-residue | 624 |
| Sequence ($|i\text{-}j| = 1$) | 341 |
| Medium-range ($|i\text{-}j| \leq 4$) | 133 |
| Long-range ($|i\text{-}j| \geq 5$) | 150 |
| Dihedral angle restraints | 134 |
| $\phi$ | 67 |
| $\varphi$ | 67 |
| **Hydrogen bond restraints [a]** | 68 |
| **Structure statistics** |  |
| Distance violations per structure |  |
| 0.1–0.2 Å | 1.8 |
| 0.2–0.5 Å | 0 |
| >0.5 Å | 0 |
| r.m.s. of distance violation per constraint | 0.02 Å |
| Max. distance violation | 0.17 Å |
| Dihedral angle violations per structure |  |
| 1–5° | 0.8 |
| >5° | 0 |
| r.m.s. of dihedral violation per constraint | 1.0° |
| Max. dihedral angle violations | 4.9° |
| **Ramachandran plot summary [b]** |  |
| Most favored regions | 90.9% |
| Additionally allowed regions | 7.5% |
| Generally allowed regions | 0.0% |
| Disallowed regions | 1.6% |
| **Averaged r.m.s.d. to the mean structure [c]** |  |
| Backbone | 0.9 Å |
| Heavy atoms | 1.4 Å |

[a] Two distance ranges (NH−O = 1.8–2.4 Å, N−O = 2.8–3.4 Å) were used for the hydrogen bond restraints between the amide and carbonyl group atoms.

[b] Ramachandran plot analysis was performed over residues 68−96, 99−121 and 123−145 with PROCHECK.

[c] Average r.m.s.d. was calculated from an ensemble of 10 structures over secondary structure elements, including α-helices: residues 68−71, 109−117 and 136−145; $3_{10}$-helix: 119−121; β-strands: 74−78, 82−87, 90−96, 100−103 and 124−126.

sorted out spectral differences in their chemical shift patterns (Fig 2C) and signal intensities (Fig 2D). We observed that more than 10 residues—including V59, V62, F91, A93, E94, G96, N99, S102, I103, K105, I123, I125 and Y126—exhibited chemical shift perturbations (CSPs) of greater than 0.06 ppm in the presence of sH2 (Fig 2C), indicating that these residues might be responsible for protein-protein interaction. Moreover, in Fig 2D, five residues displayed completely diminished signal intensities in the presence of sH2 protein (i.e., D61, N63, D68, K72 and N124) and, for a further ten residues (V62, D64, T65, I66, K70, Q71, N122, I123, Y126 and N127), signal intensities were reduced by more than 40% under the same condition.

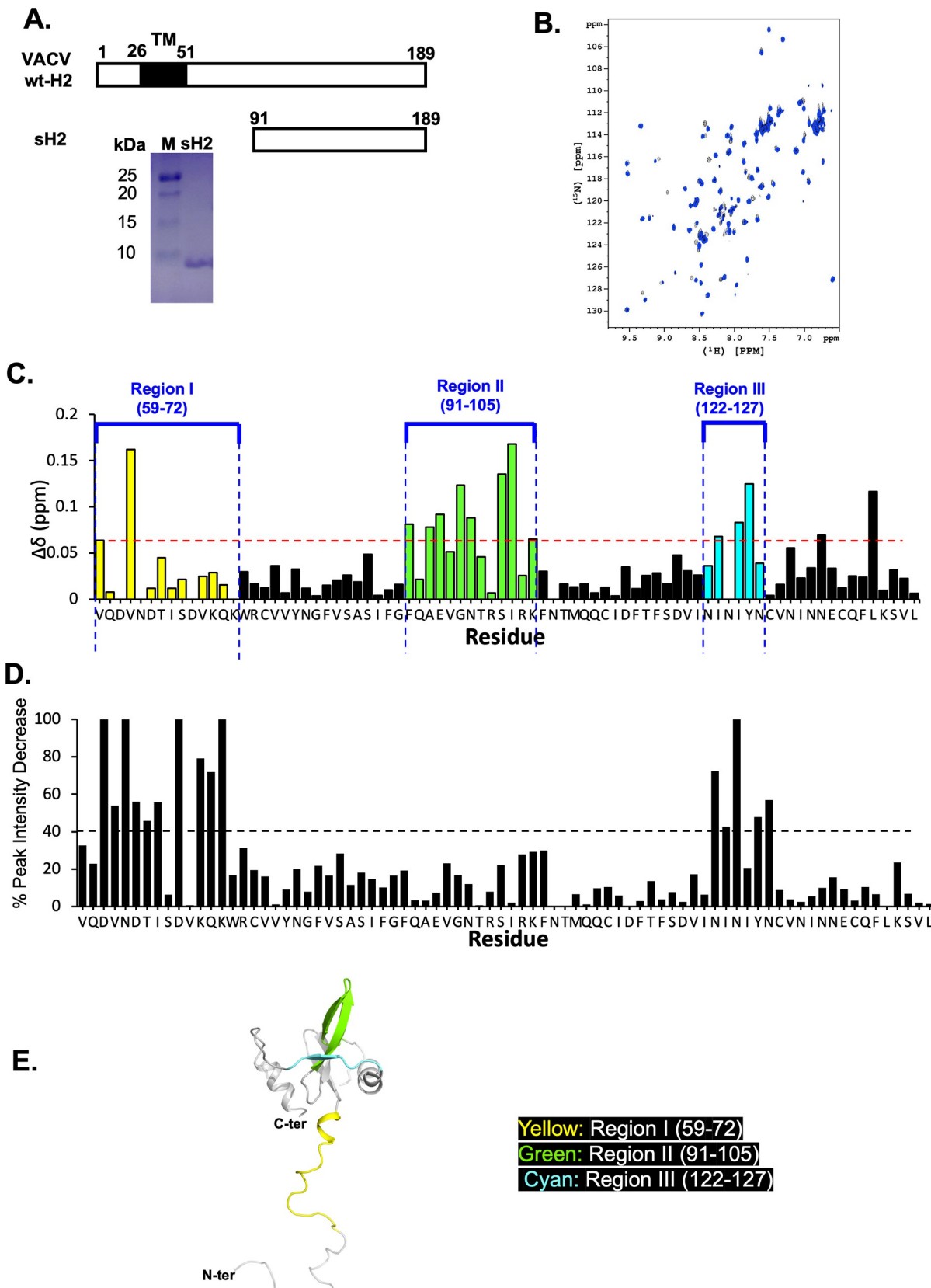

**Fig 2. VACV sA28 and sH2 *in vitro* protein binding analyses.** (**A**) Schematic representation of wild-type full-length H2 protein (aa 1–189) containing an N-terminal transmembrane domain (TM, aa 26–51). A Coomassie blue-stained SDS-PAGE gel of purified soluble sH2 (aa 91–189) is also shown. M: protein markers. (**B**) 2D $^1$H-$^{15}$N HSQC spectra of sA28 in the absence (black) and presence (blue) of sH2 at pH 6.5. The 2D HSQC amide $^1$H and $^{15}$N signals were all highly resolved and assigned to sA28 amino acids. (**C**) Chemical shift perturbations and (**D**) Signal intensity decay for sA28 amino acids due to the presence of sH2, as deduced from (B). The three most perturbed regions from the HSQC spectra are labeled with different colors: region I (aa 59–72, yellow); II (aa 91–105, green); and III (aa 122–127, cyan). The red dashed line in C marked the value of 0.06 ppm. The black dashed line in D marked 40% intensity decrease. (**E**) Ribbon representation of the NMR-derived molecular structure of tA28, with the most perturbed regions I, II and III highlighted in the respective colors shown in (C).

Based on the CSP and diminished signal intensities derived from our 2D HSQC spectra, we grouped the most perturbed residues into three regions (color-coded in Fig 2C and 2E): region I (residues 59–72, yellow); region II (91–105, green); and region III (122–127, cyan).

Next, we performed isothermal titration calorimetry (ITC) to measure the *in vitro* binding affinity between VACV sA28 and sH2 proteins (Fig 3A). In this thermodynamic binding assay, sA28 was titrated systematically to sH2 to reveal any change in enthalpy during the titration process. Through an iterative nonlinear fitting calculation, we determined the following thermodynamic parameters: binding enthalpy ($\Delta H$) -8.8 ± 0.5 kcal/mol.; binding constant ($K_D$) 4.3 ± 0.2 × 10$^{-6}$ M; and binding stoichiometry (n) 1.0 ± 0.1 (Fig 3D). Therefore, we conclude that VACV sA28 binds *in vitro* to sH2 with a moderate binding affinity at a 1:1 molar ratio. We noticed that the A28 ortholog in mpox virus, named A30, only differs from VACV A28 at 4 amino acids, i.e., Q110R, V130I, V131A and A137T, with the mpox H2 ortholog (also named H2) solely differing from VACV H2 at F185L (Fig 3B). Consequently, we performed ITC measurements on the mpox-A30 and mpox-H2 proteins (Fig 3C) and observed similar binding parameters to the respective VACV proteins, with a $K_D$ value of 5.3 ± 0.1 × 10$^{-6}$ M and a binding stoichiometry of 1:1 (Fig 3D).

## Mutagenesis analysis of VACV A28 protein to establish its functional mapping in MV infection, membrane fusion and EFC formation

To determine if the aforementioned regions I, II and III (RI, RII and RIII) of VACV A28 are important for vaccinia virus infection, we performed an *in vitro* alanine mutagenesis analysis on the A28L open-reading frame (ORF). When we performed mutagenesis designs, we postulated that surface-exposed residues would constitute domains accessible for protein-protein interaction. Therefore, we generated seven A28 mutant constructs containing changes at single or multiple surface amino acids in RI (aa59-72), three mutants in RII (aa91-105) and three mutants in RIII (aa122-127) (Fig 4A). In the full-length A28 structure modeled by AlphaFold, D114D119 appear on the protein surface close to D56R58 in RI so we generated not only the D56R58A RI mutant but also the D56R58D114D119 RI mutant. Furthermore, we picked four highly conserved residues, W73, R74, F89 and G90, that are outside of the RI, RII and RIII regions (Fig 4B) as targets to generate four conserved mutants (Fig 4A). In total, we generated 17 mutant plasmids. Next, we determined if any of these A28L mutations affected VACV infectivity using a well-established transient complementation assay system [36] with an IPTG-inducible virus (viA28) described previously [10,32].

We infected BSC40 cells with viA28 and then transfected them with individual plasmids encoding either wild type or mutant A28 protein as described in Materials and Methods. These infected and transfected (inf/tnf) cells were cultured in the absence of IPTG so that A28 protein was only expressed from the transfected plasmid. Cells were harvested at 24 hours post-infection (hpi) and then mature virus titers in the lysates were determined by plaque assay. Because EFC components are not required for post-entry steps in VACV life cycle ([7] and the references within) a reduction in MV titer would reflect a loss of vaccinia MV

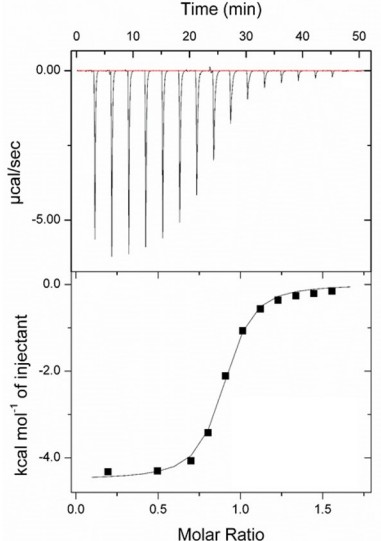

**A.**

**ITC – vaccinia sA28 vs sH2**

**B.**

**C.**

**ITC – mpox-A30 vs mpox-H2**

**D.**

| Sample | $K_D \times 10^{-6}$ (M) | n | $\Delta H$ (kcal mol$^{-1}$) | $\Delta S$ (kcal mol$^{-1}$deg$^{-1}$) x10$^{-3}$ |
|---|---|---|---|---|
| sA28 | $4.3 \pm 0.2$ | $1.0 \pm 0.1$ | $-8.8 \pm 0.5$ | $-5.0 \pm 1.9$ |
| mpox-A30* | $5.3 \pm 0.1$ | $0.9 \pm 0.1$ | $-4.3 \pm 0.3$ | $9.7 \pm 0.5$ |

**Fig 3. *In vitro* ITC analyses of vaccinia and mpox sA28 and sH2 interaction.** (**A**) Recombinant vaccinia sA28 (0.5 mM) was isothermally titrated with vaccinia sH2 (0.05 mM). The ITC exothermic heat of the reaction (upper panel) and the integrated areas of the peaks (lower panel) plotted against the molar ratio of the two proteins is shown. The best fit of the experimental data was determined from the non-linear least square fit (solid line). (**B**) Schematic drawing of vaccinia sA28 and sH2 constructs, as well as their mpox orthologs, mpox-A30 and mpox-H2. Amino acid differences between the vaccinia and mpox orthologs are indicated by red lines and labeled above the proteins. (**C**) Mpox-A30 (aa 56–146; 0.5 mM) was isothermally titrated with mpox-H2 (aa 91–189; 0.05 mM). The ITC exothermic heat of the reaction (upper panel) and the integrated areas of the peaks (lower panel) plotted against the molar ratio of the two proteins is shown. The best fit of the experimental data was determined from the non-linear least square fit using a single binding model (solid line). (**D**) Thermodynamic parameters derived from the ITC data of protein-protein interactions of (A) vaccinia A28 and H2 and (C) mpox-A30 and mpox-H2.

infectivity [10,32]. Complementation with wild type A28 plasmid increased MV titer up to ~100 fold at 24 hpi and the titer was used to represent 100% MV infectivity. As shown in Fig 5A, from the seven RI mutants and the four conserved mutants, 9 mutations resulted in various degrees of virus titer reduction, ranging from 10–75% loss. Most of the mutations did not affect A28 protein expression levels in cells, except for three mutants, $D^{56}R^{58}D^{114}D^{119}A$, $D^{68}K^{72}R^{74}A$ and $G^{90}A$, that expressed much less mutant A28 protein. However, after careful modifications of plasmid transfection conditions, as described in the Materials and Methods, to ensure that levels of these three mutant proteins were comparable to wild type A28 protein (marked by *, i.e., $D^{56}R^{58}D^{114}D^{119}A^*$, $D^{68}K^{72}R^{74}A^*$ and $G^{90}A^*$ in Fig 5A), these three mutant A28 proteins remained defective in trans-complementation assays (gray bars in Fig 5A), excluding that protein instability is the cause of reduced VACV infectivity.

**A.**

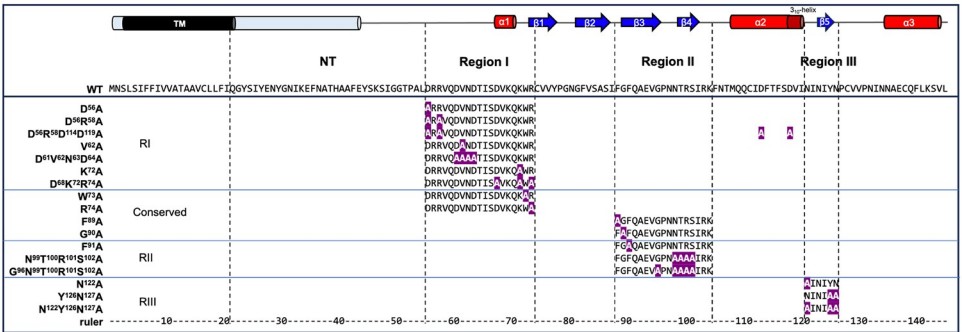

**B.**

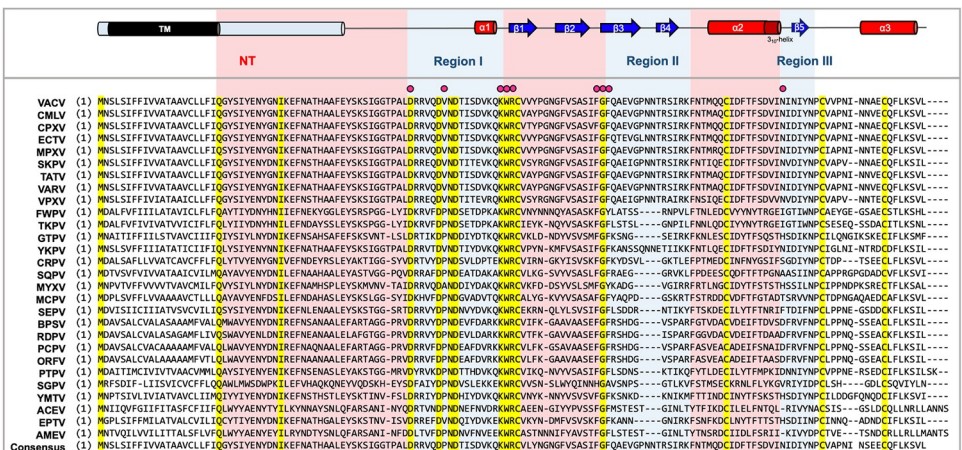

**Fig 4.** In vitro alanine mutagenesis of VACV A28L ORF (**A**) List of all designed A28 mutants, showing the regions containing single or clustered alanine mutations (the mutated alanine residues are colored in white within a purple background). The resolved and predicted secondary structures of vaccinia virus A28 and its full-length amino acid sequence are displayed on top of the list. (**B**) Multiple sequence alignment of VACV A28 protein with its orthologs from 24 representative chordopoxviruses and three entomopoxviruses. The resolved and predicted secondary structures of vaccinia virus A28 are shown above the multiple sequence alignment. NT: N-terminal region. The yellow background highlights 100% identical residues and the red dots represent the residues included in mutagenesis analyses. VACV: vaccinia virus (YP_233033.1); CMLV: camelpox virus (NP_570537.1); CPXV: cowpox virus (NP_619947.1); ECTV: ectromelia virus (NP_671649.1); MPXV: monkeypox virus (NP_536567.1); SKPV: skunkpox virus (YP_009282841.1); TATV: taterapox virus (YP_717459.1); VARV: variola virus (NP_042179.1); VPXV: volepox virus (YP_009282895.1); FWPV: fowlpox virus (NP_039155.1); TKPV: turkeypox virus (YP_009177163.1); GTPV: goatpox virus (YP_001293309.1); YKPV: yokapox virus (YP_004821485.1); CRPV: crocodilepox virus (QGT49410.1); SQPV: squirrelpox virus (YP_008658542.1); MYXV: myxoma virus (NP_051830.1); MCPV: molluscum contagiosum virus (NP_044085.1); SEPV: sea otter poxvirus (YP_009480656.1); BPSV: bovine papular stomatitis virus (NP_958014.1); RDPV: red deer parapoxvirus (YP_009112844.1); PCPV: pesudocowpox virus (YP_003457411.1); ORFV: orf virus (NP_957882.1); PTPV: pteropox virus (YP_009268838.1); SGPV: salmon gill poxvirus (AKR04251.1); YMTV: yaba monkey tumor virus (NP_938373.1); ACEV: anomala cuprea entomopoxvirus (YP_009001544.1); EPTV: eptesipox virus (YP_009408076.1); and AMEV: amsacta moorei entomopoxvirus (NP_064968.2).

Using a 50% titer reduction as cut-off, we selected five mutants—$D^{56}R^{58}D^{114}D^{119}A$, $D^{61}V^{62}N^{63}D^{64}A$, $K^{72}A$, $D^{68}K^{72}R^{74}A$ and $G^{90}A$ for further study. We first performed electron microscopy (EM) on the inf/tnf cells at 24 hpi to ensure that normal virus morphogenesis occurred in all cases and the infected cells produced abundant mutant MV particles (S2 Fig).

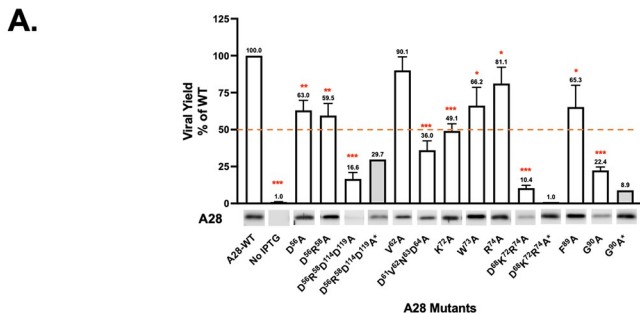

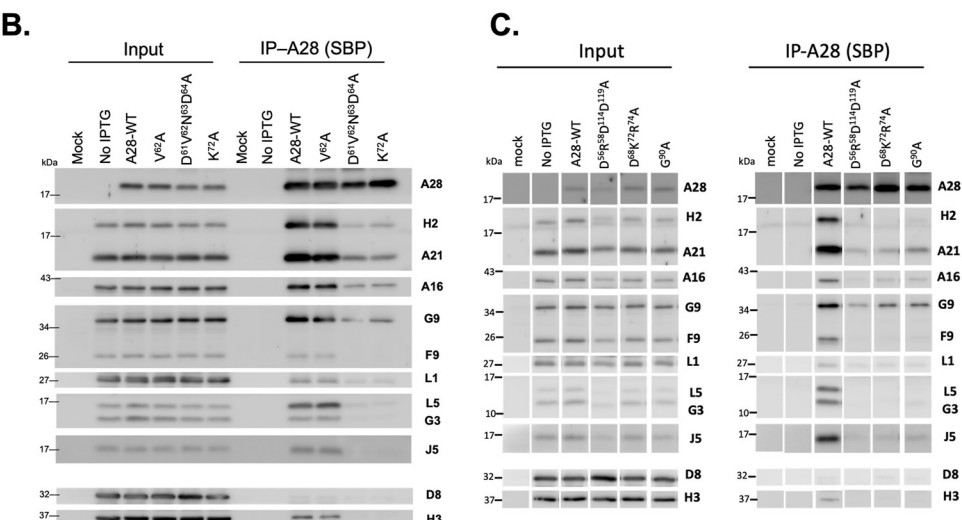

**Fig 5. Biological effects of mutations in region I (RI) and conserved residues of VACV A28.** (**A**) Virus titers obtained from transient complementation assays of A28 mutants containing mutations in region I (RI) and specific conserved residues. BSC40 cells were infected with viA28 in the absence of IPTG and transfected with either vector, wild-type (WT) or mutant A28 plasmids (RI and conserved mutants). Mutant virus yield was determined at 24 hpi by means of plaque assays and normalized to the value for WT virus. The red dashed line marks 50% virus yield on the y-axis. A28 expression in the crude lysate was detected by immunoblotting using an anti-A28 antibody. Each complementation experiments with mutant plasmids were repeated three times, except for the expression-adjusted unstable mutants ($D^{56}R^{58}D^{114}D^{119}A^*$, $D^{68}K^{72}R^{74}A^*$ and $G^{90}A^*$) that are shown as gray columns without error bars. Error bars represent standard deviations. $^*P < 0.05$, $^{**}P < 0.01$, $^{***}P < 0.001$. (**B&C**) Co-immunoprecipitation (co-IP) of A28 RI mutants was performed with streptavidin resin to capture streptavidin-binding peptide-tagged A28 and associated EFC components, as described in the Materials and Methods. The bound proteins were eluted, precipitated, and resolved in SDS-PAGE for immunoblot analysis. A function-intact mutant, $V^{62}A$, served as an additional control that co-purified other EFC components as well as WT.

We then took the inf/tnf lysates of the five mutants to perform MV-mediated membrane fusion (i.e. fusion-from-without) as described in the Materials and Methods [37]. In addition to wild type A28, the $V^{62}A$ RI mutant that also grew well was included as a control. GFP- and RFP-expressing HeLa cells were mixed at a 1:1 ratio and infected with each of the inf/tnf lysates at 37°C for 1 hour, washed with either neutral (pH7.0) or acidic (pH5.0) buffer, and then immediately monitored for MV-induced membrane fusion-from-without from 30 min to 2 hpi. As shown in the images of RI mutants in Fig 6A, at neutral pH, none of the MV in the lysates triggered fusion between GFP- and RFP-expressing HeLa cells, as expected [37]. However, at the low pH of 5.0 that mimics an endosomal acidic environment, wild type A28 and $V^{62}A$ mutant MV triggered robust membrane fusion between GFP- and RFP-expressing HeLa

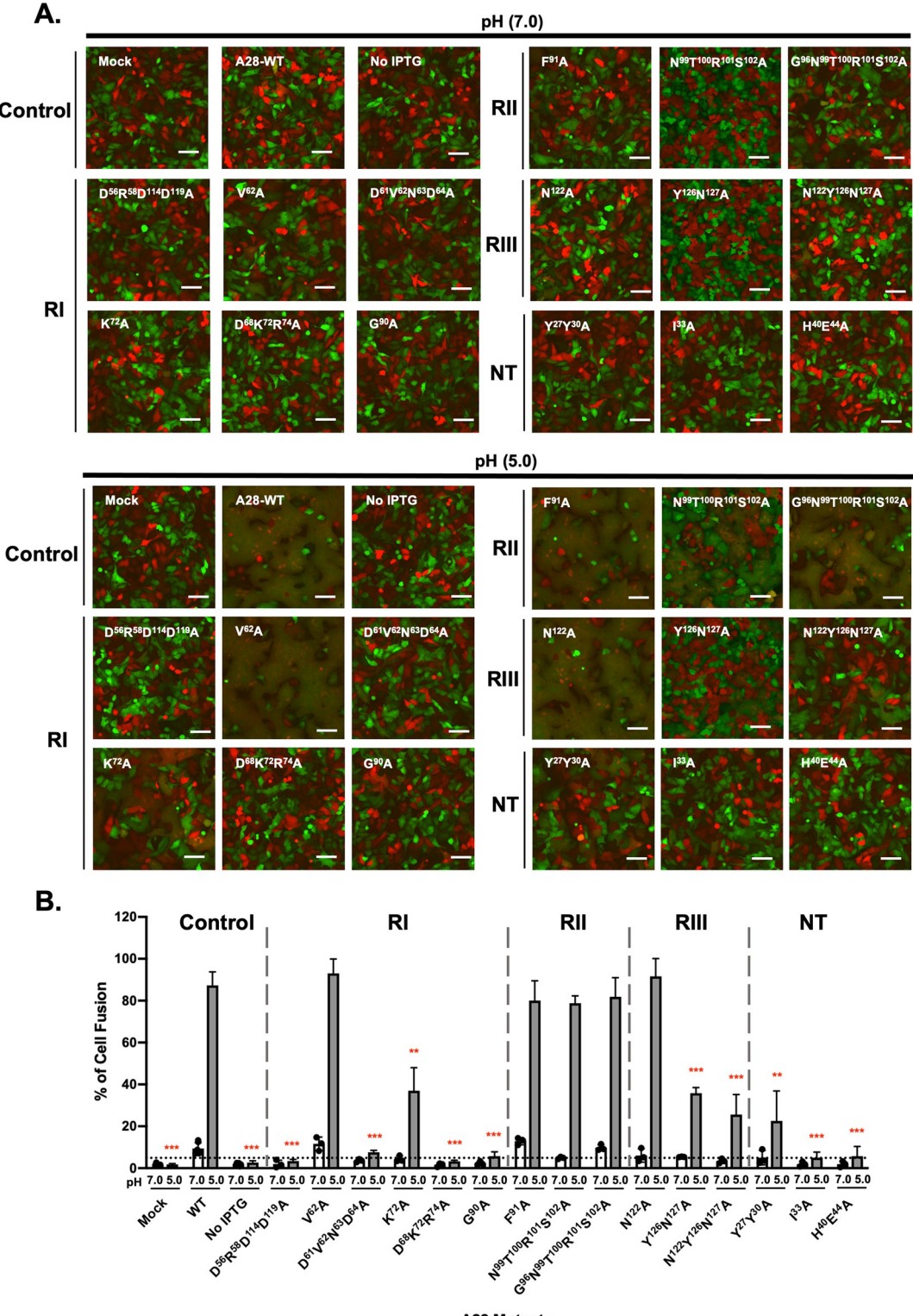

**Fig 6. MV-triggered cell fusion assay of region I, II, II and N-terminal VACV A28 mutants.** (**A**) Cell-cell fusion analyses of wild type and all the VACV A28 mutants, including RI, RII, RIII and NT mutants, at neutral pH7.0 or acidic pH5.0 as described in the Materials and Methods. An intact mutant, $V^{62}A$, served as an additional control capable of inducing fusion at low pH, just like the WT A28 protein. The images were captured at 2 hpi. Scale bar: 100 μm. (**B**) Quantification of MV-induced cell-cell fusion at pH7.0 or pH5.0 from images in A. Photographs from three independent experiments were analyzed

using Fiji software and percentages of virus-triggered cell fusion at low pH were calculated as (the surface area of GFP$^+$RFP$^+$ double-fluorescent cells divided by the surface area of single-fluorescent cells) x100%. Error bars represent the standard deviations from three independent experiments. Statistical comparisons of cell-cell fusion are between WT and each mutant at the low pH of 5.0. *$P < 0.05$, **$P < 0.01$, ***$P < 0.001$.

cells. In contrast, four RI mutants, D$^{56}$R$^{58}$D$^{114}$D$^{119}$A, D$^{61}$V$^{62}$N$^{63}$D$^{64}$A, K$^{72}$A, D$^{68}$K$^{72}$R$^{74}$A, and one conserved mutant, G$^{90}$A, displayed obvious defects in membrane fusion activity (RI in Fig 6A). Quantifications of membrane fusion were performed by measuring the percentages of GFP$^+$RFP$^+$ double-positive cells (RI in Fig 6B), as described in the Materials and Methods. Combining the results in Figs 5A, 6A&6B, the five A28 mutations that caused reduced virus titer also elicited a membrane fusion defect.

Finally, we performed coimmunoprecipitation analyses of the inf/tnf lysates and monitored the ability of the five A28 mutant proteins to interact with other EFC components. Wild type A28 (Fig 5B&5C) specifically brought down nine EFC components but not other viral membrane proteins such as D8 and H3. The control V$^{62}$A mutant also coimmunoprecipitated all the same detectable EFC components (Fig 5B). In contrast, the four RI A28 mutants—D$^{61}$V$^{62}$N$^{63}$D$^{64}$A, K$^{72}$A, D$^{56}$R$^{58}$D$^{114}$D$^{119}$A, D$^{68}$K$^{72}$R$^{74}$A and one conserved mutant G$^{90}$A (Fig 5B&5C) failed to coimmunoprecipitate H2 and other EFC components, with only trace amounts of A21 and G9 being detected in immunoblots. Taken together, our functional analysis of A28 mutant proteins revealed important amino acids in RI essential to MV infectivity, membrane fusion activity, and EFC formation, supporting the *in vitro* A28-H2 binding characterization in Fig 2. One conserved residue G90 is also important for A28 function and will be discussed below.

## Mutations in A28 regions I and III, but not II, disrupt A28-mediated MV infectivity, membrane fusion and EFC formation

We also performed transient complementation assays on three A28 RII mutants—F$^{91}$A, N$^{99}$T$^{100}$R$^{101}$S$^{102}$A and G$^{96}$N$^{99}$T$^{100}$R$^{101}$S$^{102}$A (Fig 7)—as well as three RIII mutants—N$^{122}$A, Y$^{126}$N$^{127}$A and N$^{122}$Y$^{126}$N$^{127}$A (Fig 8). Surprisingly, all three RII mutant A28 proteins remained largely functional in terms of virus yield at 24 h pi (Fig 7A), membrane fusion activity at low pH (Fig 6A&6B), and coimmunoprecipitation experiments (Fig 7B&7C), indicating that these residues in the RII domain do not play a significant role in A28-mediated EFC functions. In contrast, a triple mutation in RIII, N$^{122}$Y$^{126}$N$^{127}$A, reduced MV infectivity to less than 50%, with the N$^{122}$A single and Y$^{126}$N$^{127}$A double mutants having lesser impacts (Fig 8A). The triple mutant also displayed diminished membrane fusion ability at low pH (Fig 6A&6B) and was the least capable of forming EFC in a coimmunoprecipitation analysis (Fig 8B) when compared with the N$^{122}$A single and Y$^{126}$N$^{127}$A double mutants. Thus, collectively, our mutational analyses of RI, RII and RIII mutant A28 proteins uncovered residues in the RI and RIII regions, but not in RII, that are critical for viral EFC formation and activity.

To validate that the RI and RIII regions are required for direct binding of A28 to H2, we expressed and purified yet other soluble A28 variants (residues 56–146) containing mutations in the RI, RII and RIII regions (S3 Fig) Using gel filtration chromatography (S4A Fig) and circular-dichroism-spectra analyses (S4B Fig) these proteins exhibited homogeneous profiles. We then assayed the ability of these mutants to bind to sH2 in vitro using HSQC and/or ITC spectroscopy as described in Fig 3A. Four RI mutants—D$^{56}$R$^{58}$D$^{114}$D$^{119}$A, D$^{61}$V$^{62}$N$^{63}$D$^{64}$A, K$^{72}$A and D$^{68}$K$^{72}$R$^{74}$A (Fig 9A)—as well as one RIII mutant, N$^{122}$Y$^{126}$N$^{127}$A (Fig 9C), lost the ability to bind to sH2 *in vitro*, providing direct evidence that these specific RI and RIII residues are required for A28 and H2 protein interaction. One RI mutant V$^{62}$A (Fig 9A) and one RII

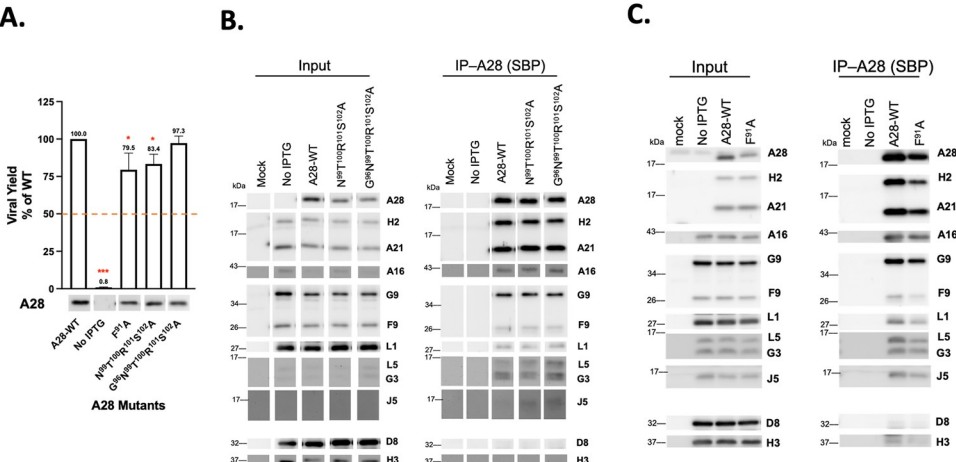

**Fig 7. Biological effects of mutations in region II of VACV A28.** (**A**) Transient complementation assay of A28 mutants containing mutations in RII revealed comparable infectivity to WT A28. BSC40 cells were infected with viA28 in the absence of IPTG and transfected with either vector, wild-type (WT) or mutant A28 plasmids (region II). Viral yields were determined at 24 hpi by plaque assays and normalized to the value for WT. The red dashed line marks 50% virus yield on the y-axis. All experiments were repeated three times. Error bars represent standard deviations. Protein expression of A28 in the crude lysate are shown below by immunoblotting with an anti-A28 antibody. $*P < 0.05$, $**P < 0.01$, $***P < 0.001$. (**B&C**) Co-immunoprecipitation was performed on A28 RII mutants using streptavidin resin to capture SBP-tagged A28 and associated EFC components, as described in the Materials and Methods. The bound proteins were eluted, precipitated, and resolved in SDS-PAGE for immunoblot analysis. A WT A28 construct was included and served as a positive control.

mutant F$^{91}$A (in Fig 9B) retained wild type A28 functions, with both those mutant proteins binding well to sH2 protein in ITC assays. All data including $K_D$ measurements for the above-described ITC experiments are summarized in Fig 9D. One sample unsuited for ITC measurement was the conserved mutant G$^{90}$A (Fig 5) because the recombinant G$^{90}$A protein formed aggregates during protein purification. It also implied that G$^{90}$A had altered the overall structure of A28 protein. Taken together, we have identified amino acid residues in the RI and RIII regions of A28 that are important for A28 and H2 interaction *in vitro*, EFC formation *in vivo*, and membrane fusion activity during VACV infections of cells (Fig 9E).

We wondered whether sA28 protein may induce a conformational change due to the sH2 binding. Since circular dichroism (CD) ellipticity at 208 and 222 nm are indicative of helical contents and that at 200 nm is sensitive to random coil in these proteins, we here measured secondary structure of sA28 (Fig 10A) and sH2 proteins (Fig 10B), respectively, as well as sA28 and sH2 protein complex (1:1 molar ratio) (Fig 10C) by CD spectroscopy. A summation of the individual CD spectrum of sA28 and sH2 protein (Fig 10D) was overlaid with the CD spectrum of the sA28 and sH2 protein complex (Fig 10E). As resolved, the sA28 and sH2 protein complex formation gave rise to a weaker molar ellipticity in the range of 208 to 222 nm and a stronger molar ellipticity close to 200 nm (Fig 10E), indicating a slightly higher random coil content and less α-helical content. Thus, our CD measurements suggested that A28 may induce a mild conformational change in the secondary structure due to the sH2 binding.

The A28 structure we solved by NMR did not contain its N-terminal region (residues 1–55), which contains a predicted hydrophobic transmembrane region (residues 1–22), a helix (residues 22–44), and a disordered region (residues 45–55). To learn more about full-length A28 protein structure, we subjected the A28 and H2 protein-protein interaction to MD simulations [38], which have been widely deployed to investigate protein-protein interactions, as well as protein interactions with lipid bilayers [39–41]. Given that certain amino acids

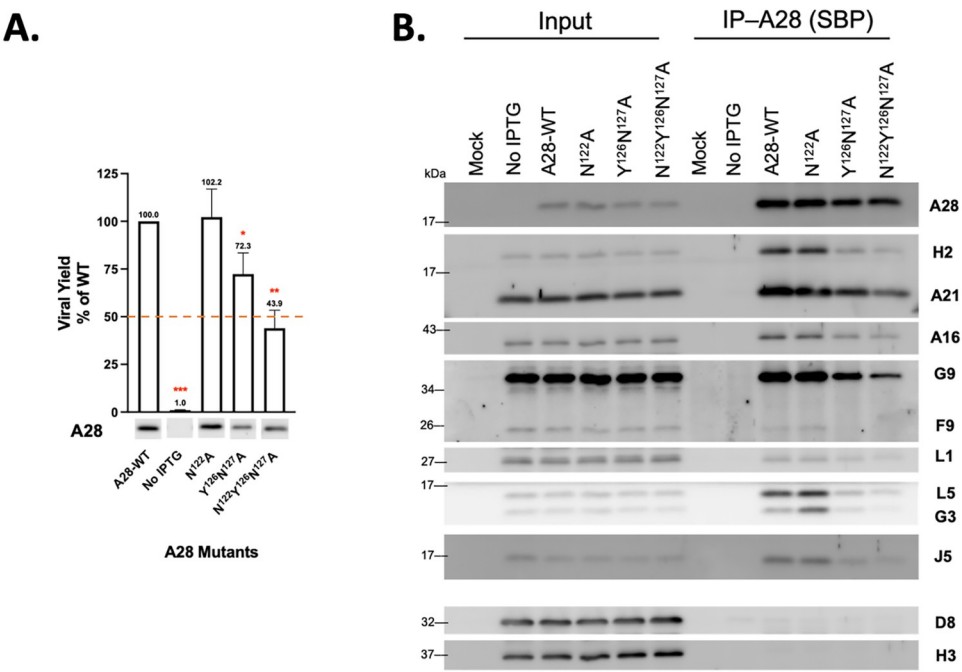

**Fig 8. Biological effects of mutations in region III (RIII) of VACV A28. (A)** Transient Virus titers obtained from transient complementation assays of A28 mutants containing mutations in region III. BSC40 cells were infected with viA28 in the absence of IPTG and transfected with either vector, wild-type (WT) or mutant A28 plasmids (RIII). Mutant virus yield was determined at 24 hpi by means of plaque assays and normalized to the value for WT virus. The red dashed line marks 50% virus yield on the y-axis. A28 expression in the crude lysate was shown below by immunoblotting using an anti-A28 antibody. All experiments were repeated three times. Error bars represent standard deviations. $^*P < 0.05$, $^{**}P < 0.01$, $^{***}P < 0.001$. **(B)** Co-immunoprecipitation was performed on the $N^{122}$A single mutant, $Y^{126}N^{127}$A double mutant and $N^{122}Y^{126}N^{127}$A triple mutant using streptavidin resin to capture SBP-tagged A28 and associated EFC components, as described in the Materials and Methods. The bound proteins were eluted, precipitated, and resolved in SDS-PAGE for immunoblot analysis. A WT A28 construct was included and served as a positive control.

throughout the entire A28 sequence are conserved in various viral strains (Fig 4B), we first used MD simulations to examine the subcomplex formation of the soluble part of A28 and the Alphafold-predicted H2 (S7 Fig), i.e., sA28 and sH2. When we compared the sA28 NMR structure (Fig 1F) and the sA28 structure in complex with sH2 as determined by MD simulation (Pink region in Fig 11A) we found that partial of α2 helix, β3 and β5 strands became random coils, suggesting that A28 undergoes a mild conformational change when interacting with H2 protein (S5 Fig). It is comforting that the MD simulation results are consistent with the CD data in Fig 10. Furthermore, MD simulations enable the identification of specific binding interfaces, as well as the residues responsible for those interactions. As shown in Fig 11A, both electrostatic and polar interactions in the region I of A28 protein were attributed to its interaction with H2 protein (residues marked in yellow in the bottom insets in Fig 11A), yet only polar interactions in region III of A28 were involved in its binding to H2 (residues marked in blue in the top insets in Fig 11A). In contrast, no A28-H2 interaction was identified in region II of the A28 protein (marked green in Fig 11B). Indeed, the residues highlighted by our MD simulations for A28-H2 interactions were in good agreement with our NMR-derived observations (Fig 2D). We also performed MD simulations with the loss-of-function RI and RIII mutants of sA28, such as $D^{61}V^{62}N^{63}D^{64}$A, $D^{68}K^{72}R^{74}$A, $K^{72}$A and $N^{122}Y^{126}N^{127}$A, to determine whether they can interact with sH2. As expected, when wild type sA28 protein formed a complex with sH2 in a total of 100ns scan length, all these sA28 mutant proteins failed to

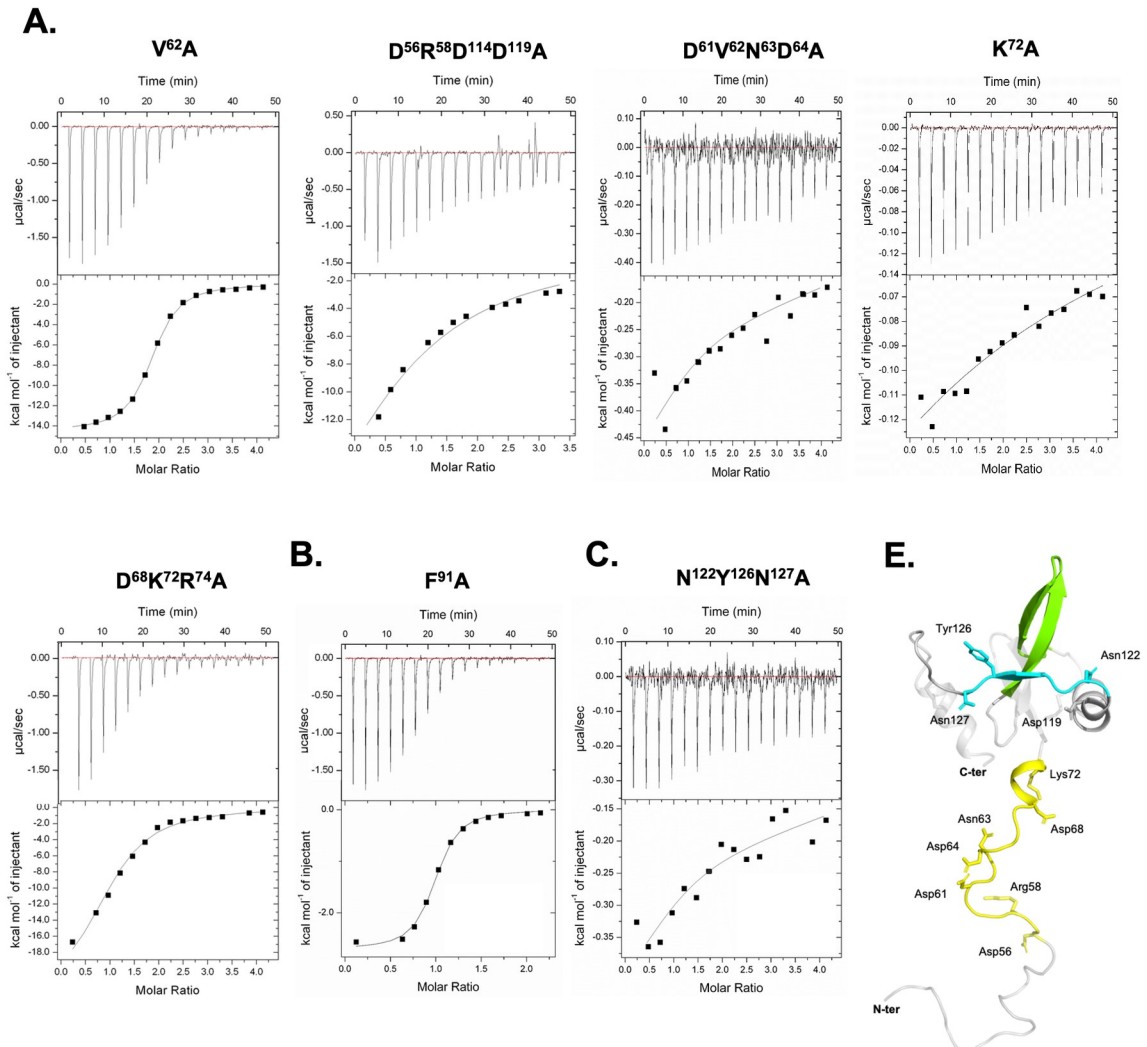

**Fig 9. *In vitro* ITC analyses for mutant sA28 protein binding to wild-type sH2 protein.** Isothermal titration calorimetry (ITC) binding profile between sA28 mutant proteins and WT-sH2 protein: (**A**) five RI mutants V$^{62}$A, D$^{56}$R$^{58}$D$^{114}$D$^{119}$A, D$^{61}$V$^{62}$N$^{63}$D$^{64}$A, K$^{72}$A and D$^{68}$K$^{72}$R$^{74}$A. (**B**) one RII mutant F$^{91}$A, and (**C**) one RIII mutant N$^{122}$Y$^{126}$N$^{127}$A. The unaffected RI mutant V$^{62}$A served as a positive control that retains its binding ability to sH2. (**D**) Summary of ITC parameters for binding of mutant sA28 to WT sH2 *in vitro*. (**E**) Ribbon representation of the NMR-derived molecular structure of tA28, in which the most perturbed regions I, II and III are highlighted in yellow, green and cyan, respectively. The amino acid residues that are functionally important in the A28-H2 interaction are individually marked with the side chains.

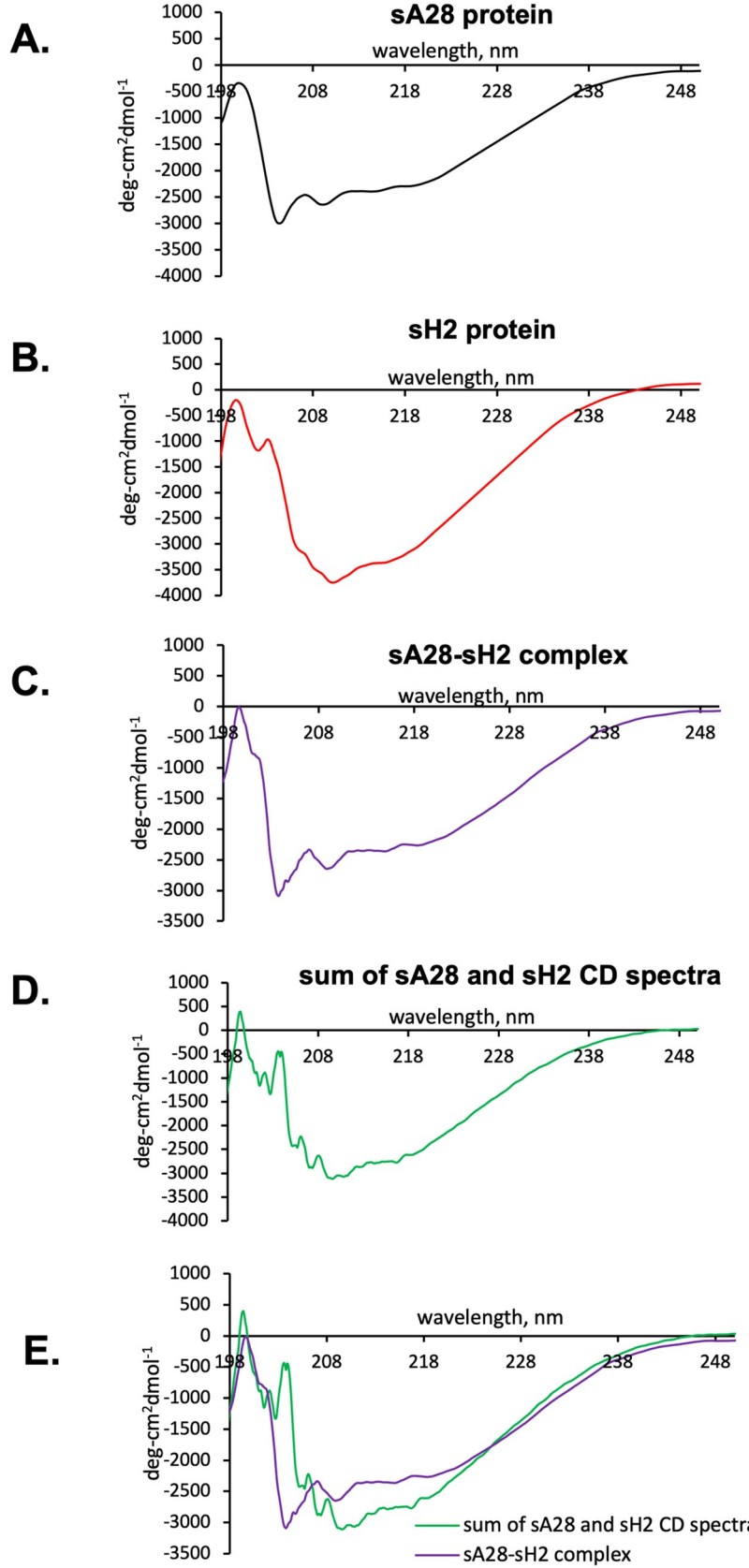

**Fig 10. Secondary structure analyses of sA28 protein when interacting with sH2 protein by circular dichroism spectroscopy.** CD spectrum of individual viral protein (**A**) sA28 and (**B**) sH2, as well as (**C**) sA28 and sH2 protein-protein complex mixed at 1:1 molar ratio, see Materials and Methods for details. (**D**) A spectral summation of the two CD spectra (**A**) and (**B**). (**E**) An overlay of (**C**) and (**D**) was presented and compared, showing that the sA28-sH2 protein complex formation gave rise to a weaker molar ellipticity in the range of 208 to 222 nm and a stronger molar ellipticity close to 200 nm. It indicated a mild conformational change in the secondary structure due to the sH2 binding.

interact with sH2 protein (S6 Fig). Finally, the MD-simulated interacting residues are supported by our experimental data from *in vitro* ITC and *in vivo* virus infection bioassays.

## Molecular dynamic (MD) simulation reveals H2 interacts with the N-terminal region of A28 (amino acids 22–55) in lipid bilayers

Given our MD simulations on the soluble A28-H2 subcomplex could be substantiated by independent experimental evidence, we deployed this approach to investigate the full-length A28 in complex with H2 and embedded in a membrane lipid bilayer, with this latter mimicking VACV membrane composition [42]. In the resulting model (Fig 12), several residues in the N-terminal region (residues 22–55) are located close to the outer membrane and are predicted to be critical for hydrophobic and polar interactions between the two viral proteins (Top insets in Fig 12). For example, our model predicts that Y27, Y30 and I33 of A28 interact with L46, A40 and T50 of H2 via hydrophobic interactions, whereas E44 and K47 of A28 interact with Q49 and D52 of H2 via polar interactions. In addition, within the A28/lipid binding interface of the inner membrane (Bottom insets in Fig 12), sidechains of Y27, Y30 and I33 of A28 interact with alkyl chains of lipid molecules, and sidechains of H40, E44 and K47 of A28 interact with acyl groups of lipid head molecules, in all cases via polar interactions. Intriguingly, our MD simulations revealed that Y27, Y30, I33, E44, and K47 of A28 may be involved in A28-H2 protein-protein interaction as well as in A28-lipid interactions.

To validate the above-mentioned residues in A28 N-terminal region (residues 22–55) to be critical for A28 function, we designed six A28 mutants- $Q^{22}A$, $Y^{27}Y^{30}A$, $I^{33}A$, $K^{34}E^{35}A$, $H^{40}E^{44}A$ and $K^{47}A$ (Fig 13A)—for biological assays as described in the previous sections for the RI, RII and RIII mutants of A28. We observed that the virus titers of three A28 mutants- $Y^{27}Y^{30}A$, $I^{33}A$ and $H^{40}E^{44}A$-were reduced by more than 50% (Fig 13A). These three mutants were also defective in acid-mediated membrane fusion activity (Fig 6A&6B). Furthermore, coimmunoprecipitation analyses revealed that these three A28 mutants brought down reduced yet detectable amounts of H2, A16 and G9 proteins (Fig 13B) but not of other EFC components, demonstrating that these residues in the N-terminal region adjacent to the N-terminal transmembrane region, are indeed biologically important for A28 function.

## Discussion

Virus entry is the first step of cell invasion, so fusion of viral and host membranes has been a topic of keen interest to biologists dissecting viral entry mechanisms in order to develop antiviral molecules that can interfere with that process. Numerous viral fusion proteins have been identified and studied over the years. The ever-increasing numbers of fusion protein structures available to scientists have allowed them to establish common pathways explaining how type I, II and III viral fusion proteins mediate membrane fusion through acid-activated conformational changes [43,44]. It is interesting to note that a recent study by Kao, et al revealed widespread distribution of poxviral EFC protein homologs in many giant DNA viruses, implying the existence of a conserved and unique membrane fusion mechanism during virus entry [45]. Nevertheless, a notable knowledge gap pertains to the large nucleocytoplasmic DNA virus

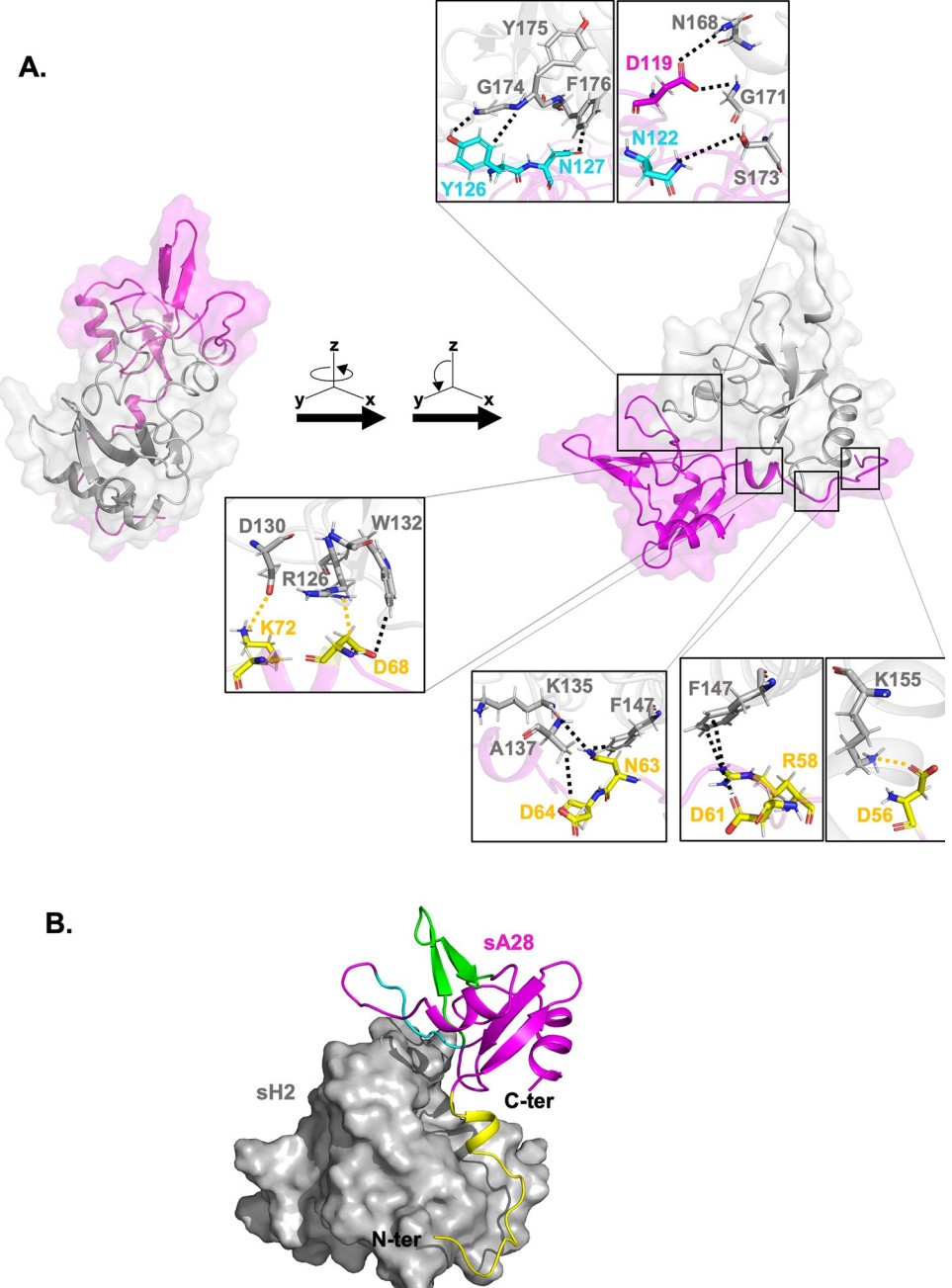

**Fig 11. MD simulations of the modeled sA28-sH2 subcomplex.** (**A**) The modeled subcomplex contains sA28 (in pink) and sH2 (in gray). Detailed molecular interactions were resolved from the binding interface and are presented in enlarged insets in the figure. The top two insets show the A28-H2 interface containing residues in RIII, i.e., D119, N122, Y126 and N127 of sA28 interacting with N168, G171, S173, G174, Y175 and F176 of sH2 via polar interactions. The four insets below illustrate A28-H2 binding interface residues in RI (yellow): D56, D68, K72 of sA28 interacting with K155, R126, D130 of sH2 via electrostatic interactions; and R58, D61, D68, N63, D64 of sA28 interacting with F147, W132, K135, A137 via polar interactions. (**B**) Shape complementarity between sA28 (in pink) and sH2 (in grey) upon complex formation. sA28 and sH2 are shown in cartoon and surface representations, respectively. The RI, II and III of sA28 are shown in yellow, green and cyan, respectively.

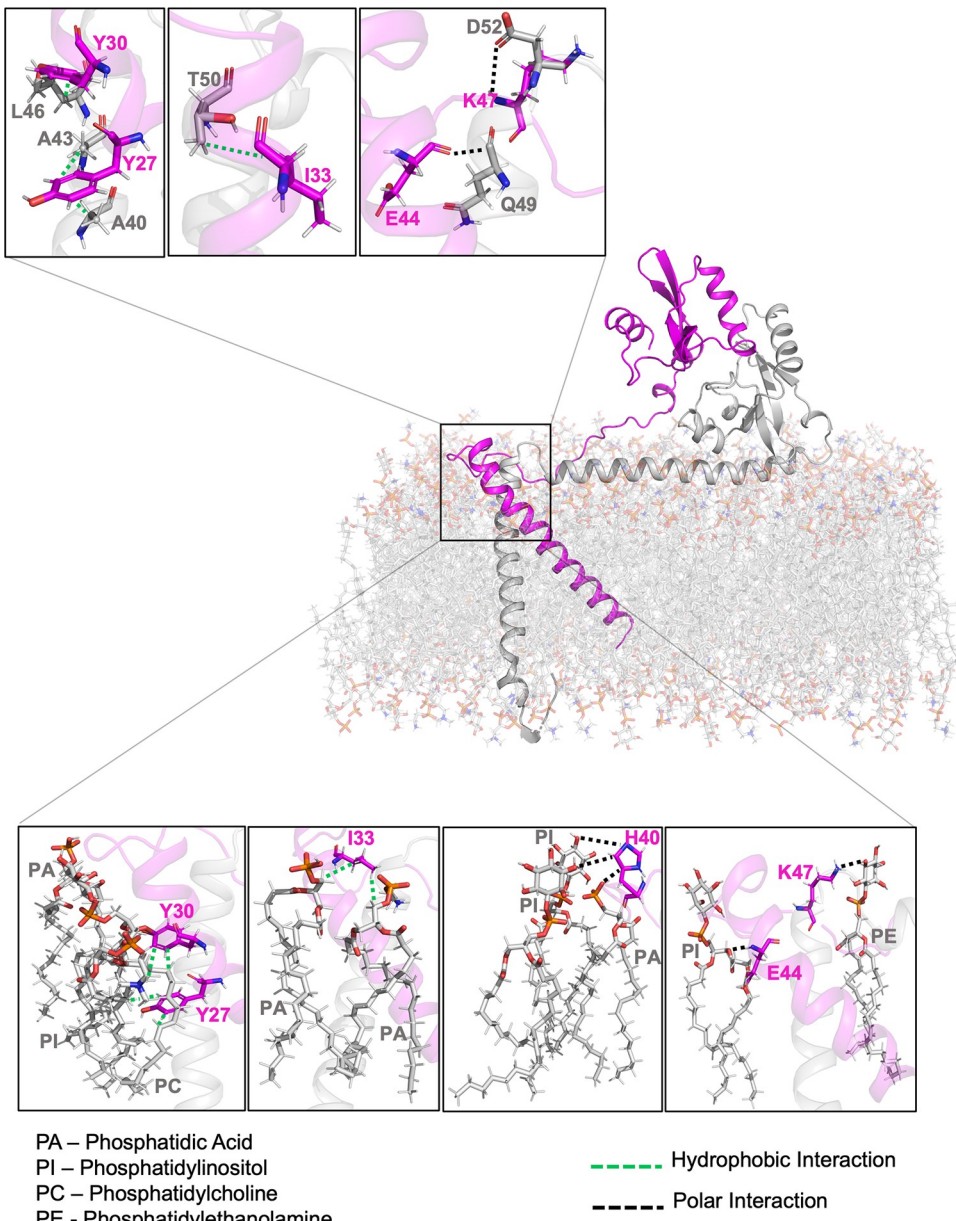

PA – Phosphatidic Acid
PI – Phosphatidylinositol
PC – Phosphatidylcholine
PE - Phosphatidylethanolamine

‒ ‒ ‒ ‒  Hydrophobic Interaction
▬ ▬ ▬ ▬  Polar Interaction

**Fig 12. MD simulation of full-length VACV A28-H2 subcomplex interacting with the viral membrane.** The MD simulation was performed for the full-length A28 (pink)-H2 (grey) subcomplex in the presence of the modeled viral lipid membrane. The lipid membrane molecules were built by mimicking VACV membrane composition [42]. The soluble C-terminal domains of both A28 and H2 were adopted from the sA28 and sH2 complex generated in Fig 10A. Protein structures of the remaining parts of A28 and H2 proteins, including the predicted TM and hydrophobic regions, were constructed using AlphaFold 2.0. Integration of the soluble and TM elements with the lipid membrane was determined using the molecular dynamics software GROMACS. The Gibbs free energies determined for the A28-lipid bilayer and A28-H2 subcomplex interactions were -2.4 and -0.4 kcal/mol, respectively, indicating that the molecular interaction of the former is rather stable. Top three enlarged insets showed that five residues of A28 protein, Y27, Y30, I33, E44 and K47, are positioned at the H2 binding interface in the presence of the viral lipid bilayer (light gray), with Y27, Y30 and I33 of A28 interacting with L46, A40 and T50 of H2 via hydrophobic interactions; and E44, K47 of A28 interacting with Q49 and D52 of H2 via polar interactions. Moreover, the lower four panels showed that these five residues also interact with membrane lipids, i.e. the sidechains of Y27, Y30 and I33 of A28 interact with the alkyl chain of lipid molecules, whereas the sidechains of H40, E44 and K47 exert polar interactions with carboxyl groups of the lipid head molecules.

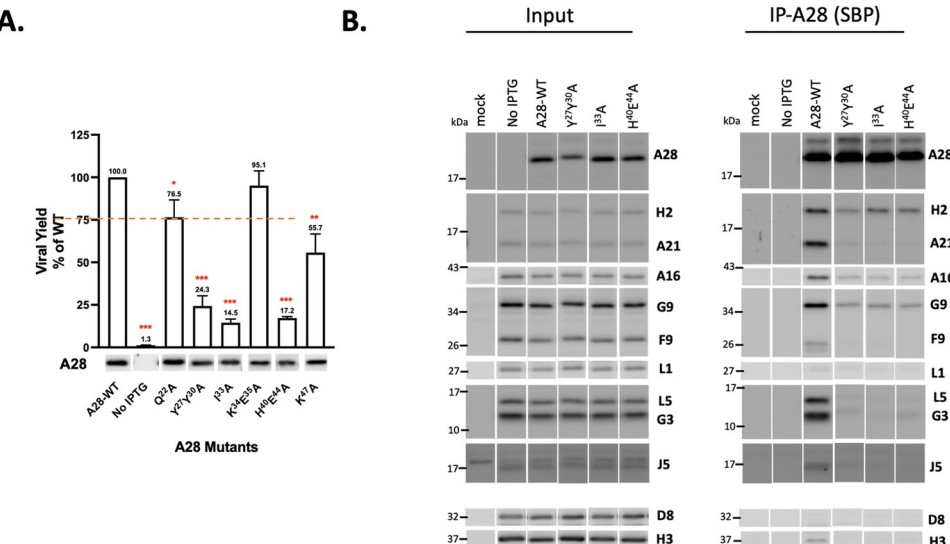

**Fig 13. Biological effects of mutations in the N-terminal region (NT; aa 22–55) of VACV A28.** (**A**) Transient complementation assay of A28 N-terminus mutants reveals residues important for viral infectivity. Viral yield was normalized to that of wild-type (WT) by dividing the titer at 24 hpi of each mutant with of the value for WT. The red dashed line marks 50% virus yield on the y-axis. A28 expression in the crude lysate, shown at the bottom, was detected by immunoblotting using an anti-A28 antibody. All experiments were repeated three times. Error bars represent standard deviations. *$P < 0.05$, **$P < 0.01$, ***$P < 0.001$. (**B**) Co-immunoprecipitation was performed on A28 N-terminal mutants using streptavidin resin to capture SBP-tagged A28 and associated EFC components, as described in the Materials and Methods. The bound proteins were eluted, precipitated, and resolved in SDS-PAGE for immunoblot analysis. A WT A28 construct was included and served as a positive control.

family that includes poxviruses. Poxvirus-mediated membrane fusion has primarily been studied in VACV EFC consisting of 11 protein components dedicated to membrane fusion during MV and EV entry [6,7,46]. Genetic data from knockout mutants support that the EFC plays a critical role in poxvirus-mediated membrane fusion. Crystal structures of the L1 and F9 components were reported previously, revealing structural similarities that imply a shared evolutionary origin [12,47]. More recently, crystal structures of truncated G3/L5 [27] and G9/A16 [28] heterodimers have also been reported. Although individual subcomponent structures do not provide an overall picture of EFC assembly, they provide important insights that can inform future structural determinations of the entire EFC.

In this study, we obtained an NMR structure of vaccinia tA28 (residues 38–146) protein. The NMR structure of soluble tA28 revealed an α-β-α sandwich-like structural motif from residues 68–146, in which the β-strands are stabilized by multiple hydrogen bonds among strands sandwiched by two α-helices, whereas residues 38–67 were uncovered as a disordered domain. Recently, the protein structure prediction program AlphaFold demonstrated high accuracy in modeling 3D protein structures based on linear amino acid sequences [48,49]. Indeed, AlphaFold-generated predictions for structured domains in sA28 (56–146) (S7 Fig) are very similar to those determined by solution NMR spectroscopy, except that the predicted folded structure is not as compact as the NMR-defined structure in solution. In addition, AlphaFold predicted that A28 residues $D^{119}V^{120}I^{121}$ act as a flexible loop in a turn-like conformation (S7 Fig), rather than the short $3_{10}$-helix resolved from the NMR structure (Fig 1F).

During protein purification, we noticed that, although tA28 and sA28 proteins both were quite soluble, tA28 tended to form oligomers at a high protein concentration (i.e., 0.1 mM) or after longer purification times (i.e., 2 weeks at 4°C). Accordingly, to investigate A28 interaction with H2, it was more appropriate to use sA28 which was $^{15}$N-labeled while adding

unlabeled sH2 protein in 2D HSQC analysis. This analysis revealed that sA28 and sH2 bind *in vitro* with intermediate affinity within the μM range and with a 1:1 stoichiometry. We also investigated mpox-A30 and mpox-H2 interactions *in vitro* and, as expected, recovered binding data similar to that obtained for VACV A28 and H2.

Although our 2D HSQC analyses indicated that RI, II and III of VACV A28 might be involved in binding to sH2, our *in vitro* alanine mutagenesis strategy, encompassing a total of 17 mutations, revealed that only RI and III are important for the A28-H2 interaction, both *in vivo* and *in vitro*. One possible explanation for this discrepancy is that the chemical shift for RII in the 2D HSQC data (Fig 2C) may be an indirect consequence of A28-H2 subcomplex formation rather than contributing to A28-H2 interaction. Instead, diminished signal intensities derived from the 2D HSQC spectra (Fig 2D) predicted better for A28-H2 binding regions. All the A28 RI and RIII mutants that failed to bind H2 and form EFC exhibited low virus infectivity and a deficiency in membrane fusion at low pH, implying that the A28 and H2 interaction could be a rate-limiting step during EFC assembly. It is interesting to point out that although D119 is not in Region III, it forms close contact with F115 and N122 that in turn contact closely with N124 to help stabilize the short β-strand ranging from N124 to Y126 in Region III (122–127) of A28 protein (S8 Fig). The G90 residue at the β3 strand forms H-bonding with S87 at the β2 strand and Y126 on the β5 strand, critical for stabilizing β2, β3, and β5 secondary structures whereas G$^{90}$A disrupts H-bonding leading to a collapse of these secondary structures, therefore disfavors H2 interaction. Such structural disruption by G$^{90}$A mutation also explained our observation that the recombinant A28$^{G90A}$ mutant protein was prone to aggregation during protein purification.

Due to certain limitations of our NMR structure, we also conducted MD simulations to acquire additional structural information on VACV A28. To reconstitute the A28-H2 subcomplex in viral membrane, we generated a computer model of full-length A28-H2 subcomplex interacting with membrane molecules. In our computer simulation, the full-length A28 protein structure is based on our NMR dataset (residues 56–146) combined with the MD-simulated structure of the N-terminal region (residues 1–55), which includes the predicted transmembrane region (residues 1–22). To mimic the lipid membrane of VACV MV and minimize energy calculations, we modeled the viral membrane in terms of the composition of its lipid components [42]. Although MD simulation is not the same as experimentally determining crystal structures, the A28-H2 subcomplex structures we modeled are largely supported by our mutational analysis data. Although we did not design mutations in the C-terminal region of A28 protein, Turner, et al previously studied vaccinia virus temperature-sensitive mutants and showed that A28 truncation of 14 aa at the C-terminus, A28$^{(\Delta 133-146)}$, remained functional at 31˚C but not at 39˚C [11]. From the structural point of view, A28$^{(\Delta 133-146)}$ lacked the α3 segment, resulting in disrupting the α-β-α sandwich structure and destabilizing the β-strands. Besides, one pair of disulfide bonds between C129-C139 is also missing in A28$^{(\Delta 133-146)}$. Thus, even though A28$^{(\Delta 133-146)}$ contains intact H2-binding domains and remains functional at 31˚C, the above-mentioned structure changes caused thermodynamic instability at a higher temperature and, consequently, a loss of function. On the other hand, we sequenced a recombinant vaccinia virus WR32-7/Ind14K [50,51] (also named as IA27L [52]) that contains an A28L ORF encoding a longer A28 protein with an extra 10 amino acids fused at the C-terminus (S9 Fig) which apparently did not interfere with virus infectivity.

We want to emphasize that knowing the structure and function relationship of individual EFC components is the first step in understanding how the EFC functions. For example, anti-A28 antibodies were reported to recognize an epitope between residues 73–92 of A28 protein and neutralize vaccinia virus infection [53]. Based on our NMR structure of tA28 protein, this neutralizing epitope covers β1, β2 and a portion of the β3 strand sandwiched between the RI

and RII regions. Therefore, it is possible that when these antibodies bind to A28 protein, they interfere with H2 binding to A28 protein through the steric hindrance [53]. A previous study showed that the A28-H2 complex formation enhances the neutralizing antibody response raised by A28 protein alone [54], presumably by stabilizing an immunogenic conformation of A28. Although we only observed minor structural alteration of sA28 protein upon its binding to sH2 in CD analysis (Fig 10) and MD simulation (S5 Fig), these observations may imply a possibility that the A28-H2 interaction facilitates the exposure of neutralizing epitope(s). In the future, we will continue to investigate the structure and functional relationships of other VACV EFC components that will yield important and complementary datasets to gain an overall understanding of how each component functions in vaccinia viral EFC assembly. Previous studies suggested that A28 protein is essential for viral EFC formation [6,10] and that mutant vaccinia MV particles devoid of A28 protein can initiate hemifusion between host and viral membranes [25]. Our A28 mutagenesis results appeared consistent with that idea since all the A28 mutants that have lost MV infectivity are also defective in EFC assembly. Taken together, these data imply that a stable viral EFC structure, but not the membrane fusion step, requires A28 protein. Since vaccinia EFC contains 11 component proteins presumably one or more of the other EFC proteins may directly participate in the membrane fusion step. Clearly, identifying which component(s) of vaccinia EFC serves as the canonical fusion protein(s) to initiate membrane fusion will be the most exciting research direction in the coming years. One may anticipate that mutations of a candidate viral fusion-genic domain in EFC will lead to a fusion-defective phenotype without interrupting viral EFC formation. Although the workload is high and time-consuming, the structure and functional investigation of each viral EFC component protein will lay out an important foundation crucial for dissecting not only the viral EFC assembly but also the membrane fusion mechanism of VACV entry.

## Materials and methods

### Soluble VACV and mpox A30/H2 plasmid construction, protein expression and purification

The vaccinia sA28 (residues 56–146 a.a) and sH2 (residues 91–189 a.a) plasmids used for protein expressions were previously constructed using vaccinia A28L and H2 ORF cloned into a pET28a-SUMO plasmid vector. The mpox sA30 and sH2 expression plasmids were generated from the above vaccinia sA28 and sH2 constructs by site-directed mutagenesis using the Quik-Change Lightning Site-Directed Mutagenesis Kit (Agilent). All the vaccinia A28 mutant constructs, $D^{56}R^{58}D^{114}D^{119}A$, $D^{61}V^{62}N^{63}D^{64}A$, $K^{72}A$, $D^{68}K^{72}R^{74}A$, $N^{122}Y^{126}N^{127}A$, $V^{62}A$ and $F^{91}A$, were also constructed by site-directed mutagenesis. All the mutations were confirmed by sequencing (Genomics, Inc. Taiwan).

E.coli BL21(DE3) competent cells were transformed with individual expression plasmids encoding tA28, sA28, sH2, and mpox-A30 and mpox-H2 protein. Single colonies were cultured at 37°C for 14–16 hrs in 10 ml LB media containing 100 µg/ml kanamycin and then inoculated into 1 liter cultures and shaken at 37°C until the $OD_{600}$ reached 0.5–0.9. Isopropyl β-D-1-thiogalactopyranoside (IPTG) was added to the cultures at a final concentration of 0.05 mM and continued incubation at 17°C. Cells were collected by centrifuging at 5,000 rpm for 40 minutes at 4°C. Cell pellets were suspended in 20 mM Tris (pH 8.0) buffer with 280 mM NaCl and 10 mM imidazole (pH 8.0). Lysozyme was added to all samples to help quickly break the cells. Sonication was done using short bursts of 3-sec pulse for 10 min followed by 5 min cooling. Sonication was repeated three times in an ice bath. After lysis, the samples were centrifuged for 1 hr at 4°C to remove cell debris. Supernatants were incubated with a pre-packed Ni-NTA resin in a PD10 column. Flow-through was collected, and columns were washed

three times with 10 ml of 20 mM Tris (pH8.0) with 20 mM imidazole (pH8.0). All protein samples were eluted with the same buffer containing 500 mM Imidazole. Purified protein samples were treated with SUMO protease to cleave off tags overnight at 4˚C and further purified through size-exclusion chromatography (SEC) using a Sephadex 75 size-exclusion column on an Agilent system. The purity of each sample was confirmed using sodium dodecyl sulfate-polyacrylamide (15%) gel electrophoresis (SDS-PAGE).

When the protein samples were prepared for NMR analysis the bacteria were cultured in M9 media containing D-[$^{13}$C] glucose (U-$^{13}$C$_6$, 99%, CLM-1396-1, Cambridge Isotope) and $^{15}$NH$_4$Cl ($^{15}$NH$_4$ Chloride, 98 atom % $^{15}$N, Sigma Aldrich) as carbon and nitrogen sources in order to produce proteins uniformly labeled with these stable isotopes.

## NMR spectroscopy and structural determination

All NMR spectra were recorded at 298 K on a Bruker AVANCE 800 MHz spectrometer equipped with a 5 mm triple resonance TXI cryogenic probe. Samples of 1.5 mM $^{15}$N, $^{13}$C-labeled tA28 in 50 mM NaCl, 50 mM MES, 90% H$_2$O/10% D$_2$O at pH 6.5 were loaded into 5 mm Shigemi NMR tubes for NMR experiments. Sequential backbone and sidechain resonance assignments were as reported previously [33], and the $^1$H, $^{13}$C, and $^{15}$N chemical shift data have been deposited in the Biological Magnetic Resonance Databank (BMRB) as entry 50469.

NOE restraints were retrieved from 3D $^{13}$C- and $^{15}$N-edited NOESY-HSQC spectra with a mixing time of 150 ms. Assignments of NOE resonances were performed using NMRView 9.2.0 software, with NOE crosspeaks being categorized as very weak, weak, medium or strong according to their intensities. Hydrogen bond restraints were identified according to the slow exchange of H$^N$ signals with the solvent D$_2$O. Dihedral angle restraints Φ and Ψ were predicted from chemical shift data using the TALOS+ webserver [55]. For the final set of protein structure calculations, 100 structures of tA28 were generated with the XPLOR-NIH 3.4 software [56,57] using a standard simulated annealing protocol, i.e., starting from high temperature dynamics at 1000 K and then cooling to 100 K. Twenty structures with the lowest energy were selected for water refinement using the AMPS-NMR web portal [58] and a standard restrained molecular dynamics protocol implemented within the AMBER99SB-ILDN force field with a generalized Born model, ionic strength of 50 mM NaCl, and a 10 Å TIP3P water box. The final ensemble of 10 tA28 structures with no distance or dihedral angle restraints greater than 0.5 Å or 5˚, respectively, was selected based on their conformational energies. The quality of the final ensemble was assessed and validated using the Protein Structure Validation Suite (PSVS) [59]. Protein structure figures were prepared with the Chimera 1.14 graphics program [60]. Statistics for the resulting structure are summarized in Table 1. The 10 conformers of tA28 have been deposited in the Protein Data Bank with a PDB ID code 8GQO.

## ITC analysis of A28-H2 interaction

Calorimetric measurements were performed for wild-type and mutant sA28-tH2 interactions using a MicroCal ITC200 system (Malvern). All experiments were conducted in a buffer of 20 mM MES with 50 mM NaCl pH 6.5 at 25˚C with a stirring speed of 300 rpm to minimize protein precipitation. Before initiating the runs, analyte sH2 (0.05 mM) and ligand sA28 (0.5 mM) samples were first degassed for 10 minutes and filtered using 0.2-micron sterile filters. A total of 18 injections (2 μl ligand) were done for each run, and a 150-second time interval was set between injections. All data analysis and peak integration were conducted using the MicroCal ITC analysis software Origin 7 (Malvern).

## Circular dichroism spectra analyses of A28-H2 interaction

Circular dichroism analysis was performed to help identify any secondary structure perturbations occurring upon sA28 and sH2 binding. The CD spectrum of recombinant proteins sA28 (50 μM), sH2 (50 μM), and sA28-sH2 (1:1 mole ratio, 50 μM) complex samples in 20mM MES/50mM NaCl at pH6.5 were recorded over the wavelength range from 250 to 190 nm, in 0.1 nm steps, as an average of 10 accumulations, using a 1 mm path-length quartz cuvette on a Jasco-815 spectrometer (Jasco Inc., Japan). All CD analyses were done at 25˚C.

## MD simulations of A28-H2 interaction with viral membrane

The A28-H2 subcomplex structure based on the A28 NMR structure and AlphaFold predictions was first prepared, before selecting lipid components to build the viral membrane structure in the online software CHARMM-GUI, with this latter also used to generate the initial model of the A28-H2 subcomplex inserted into the membrane. This initial model was subsequently uploaded to the GROMACS package [61,62] for initial energy minimization, equilibration to temperature (298 K) and pressure (1 atm), and finally subjected to MD simulation for 200 ns until the system had reached minimum energy and an equilibrium state. Details of the MD simulations are described in detail below.

For the A28-H2 interaction, one molecule of A28 and one molecule of H2 were packed into a cubic box with dimensions ~300 Å using the software PACKMOL [63] to establish the initial point for the MD simulations. This step ensured that no repulsive interactions would disrupt or cause an error during the simulations. Using a V-rescale thermostat, the overall temperature of water and protein molecules was kept constant by coupling each group of molecules independently at 300 K. A Parrinello-Rahman barostat [64] was used to couple the pressure to 1 atm in every dimension separately. The time constants for the temperature and pressure couplings were 0.1 and 2 ps, respectively. A time step of 2 fs was achieved using the leapfrog algorithm to integrate equations of motion for the system. Periodic boundary conditions were also set for the whole system. We set a 1 nm cut-off for the Lennard–Jones and Ewald summation of Coulombic interactions. The Fourier space element of Ewald splitting was calculated using the Particle-Mesh-Ewald method [65] by applying cubic spline interpolation and a 0.16 nm grid length on the side. A TIP3P water model [66] was used, and the protein parameters were obtained from the AMBERff99SB-ILDN force field [67]. MD simulations were done for a total of 100 ns scan length.

## Cell cultures and viruses

BSC40, GFP- and RFP-expressing Hela cells were cultured in Dulbecco's modified Eagle's medium (DMEM) supplemented with 10% fetal bovine serum (FBS), 100 units/ml of penicillin and 100 μg/ml of streptomycin (Invitrogen). The A28-inducible virus, viA28, kindly provided by Bernard Moss, expressed A28L-HA protein in the presence of 100 μM isopropyl-β-d- thiogalactopyranoside (IPTG); the viA28 virus stock was prepared as described previously [10,32].

## Construction of A28 expression plasmids and site-directed mutagenesis

The NCBI reference sequence of VACV WR strain (NC_006998.1) contains D124 in the A28L translated protein sequences but the VACV WR strain in our laboratory contains an N124 in the translated A28 protein sequences, same as other reported VACV WR strains [11,32], VACV Copenhagen strain [2], Lister strain [68] and MVA strain [69]. A wild-type A28L expression plasmid was constructed by inserting the VACV p11k promoter upstream of the VACV WR strain A28L ORF, which was subsequently fused in-frame with a streptavidin-

binding-peptide tag (SBP, MDEKTTGWRGGHVVEGLAGELEQLRARLEHHPQGQREP) at the 3' end, resulting in pCRII-TOPO-A28-SBP. All of the A28L mutant constructs were prepared using a QuikChange Lightning Site-Directed Mutagenesis Kit (Agilent) according to the manufacturer's instructions and confirmed by sequencing (Genomics Inc., Taiwan). The primers for *in vitro* alanine mutagenesis are listed below:

| Mutant | Region | Primers used for mutagenesis |
|---|---|---|
| $Q^{22}A$ | NT | Forward: TGTGTTTACTTTTTATCGCTGGTTACTCAATATATGAA<br>Reverse: TTCATATATTGAGTAACCAGCGATAAAAAGTAAACACA |
| $Y^{27}Y^{30}A$ | NT | Forward: ATCCAGGGTTACTCAATAGCTGAAAATGCTGGCAATATTAAGGAATTT<br>Reverse: AAATTCCTTAATATTGCCAGCATTTTCAGCTATTGAGTAACCCTGGAT |
| $I^{33}A$ | NT | Forward: TATGAAAATTATGGCAATGCTAAGGAATTTAATGCTACT<br>Reverse: AGTAGCATTAAATTCCTTAGCATTGCCATAATTTTCATA |
| $K^{34}E^{35}A$ | NT | Forward: GAAAATTATGGCAATATTGCTGCATTTAATGCTACTCATGC<br>Reverse: GCATGAGTAGCATTAAATGCAGCAATATTGCCATAATTTTC |
| $H^{40}E^{44}A$ | NT | Forward: GGAATTTAATGCTACTGCTGCAGCATTCGCATATTCAAAATCTATAGG<br>Reverse: CCTATAGATTTTGAATATGCGAATGCTGCAGCAGTAGCATTAAATTCC |
| $K^{47}A$ | NT | Forward: GCAGCATTCGAATATTCAGCATCTATAGGTGGAACACCG<br>Reverse: CGGTGTTCCACCTATAGATGCTGAATATTCGAATGCTGC |
| $D^{56}A$ | Region I | Forward: CGGTGTTCCACCTATAGATGCTGAATATTCGAATGCTGC<br>Reverse: GACATCTTGAACTCTCCTAGCTAATGCCGGTGTTCC |
| $D^{56}R^{58}A$ | Region I | Forward: GGAACACCGGCATTAGCTAGGGCAGTTCAAGATGTCAACGAC<br>Reverse: GTCGTTGACATCTTGAACTGCCCTAGCTAATGCCGGTGTTCC |
| $D^{56}R^{58}D^{114}D^{119}A$ | Region I | Forward: CGATGCAACAATGTATAGCCTTTACATTTTCTGCTGTTATTAACATCAATATT<br>Reverse: AATATTGATGTTAATAACAGCAGAAAATGTAAAGGCTATACATTGTTGCATCG |
| $V^{62}A$ | Region I | Forward: GATAGGAGAGTTCAAGATGCTAACGACACAATTTCTGATG<br>Reverse: CATCAGAAATTGTGTCGTTAGCATCTTGAACTCTCCTAT |
| $D^{61}V^{62}N^{63}D^{64}A$ | Region I | Forward: CATTAGATAGGAGAGTTCAAGCTGCAGCTGCAACAATTTCTGATGTAAAGC<br>Reverse: GCTTTACATCAGAAATTGTTGCAGCTGCAGCTTGAACTCTCCTATCTAATG |
| $K^{72}A$ | Region I | Forward: ATTTCTGATGTAAAGCAAGCTTGGAGATGTGTGGTTTAT<br>Reverse: ATAAACCACACATCTCCAAGCTTGCTTTACATCAGAAAT |
| $D^{68}K^{72}R^{74}A$ | Region I | Forward: CGACACAATTTCTGCTGTAAAGCAAGCTTGGGCATGTGTGGTTTATCCAG<br>Reverse: CTGGATAAACCACACATGCCCAAGCTTGCTTTACAGCAGAAATTGTGTCG |
| $W^{73}A$ | Conserved | Forward: TCTGATGTAAAGCAAAAGGCCAGATGTGTGGTTTATCCA<br>Reverse: TGGATAAACCACACATCTGGCCTTTTGCTTTACATCAGA |
| $R^{74}A$ | Conserved | Forward: GATGTAAAGCAAAAGTGGGCATGTGTGGTTTATCCAG<br>Reverse: CTGGATAAACCACACATGCCCACTTTTGCTTTACATC |
| $F^{89}A$ | Conserved | Forward: TTTGTATCCGCTTCCATAGCCGGATTTCAGGCAGAAGTT<br>Reverse: AACTTCTGCCTGAAATCCGGCTATGGAAGCGGATACAAA |
| $G^{90}A$ | Conserved | Forward: GTATCCGCTTCCATATTGCCTTTCAGGCAGAAGTTGG<br>Reverse: CCAACTTCTGCCTGAAAGGCAAATATGGAAGCGGATAC |
| $F^{91}A$ | Region II | Forward: TCCGCTTCCATATTTGGAGCTCAGGCAGAAGTTGGACC<br>Reverse: GGTCCAACTTCTGCCTGAGCTCCAAATATGGAAGCGGA |
| $N^{99}T^{100}R^{101}S^{102}A$ | Region II | Forward: GCAGAAGTTGGACCCAATGCTGCAGCTGCAATTAGAAAATTTAACACG<br>Reverse: CGTGTTAAATTTTCTAATTGCAGCTGCAGCATTGGGTCCAACTTCTGC |
| $G^{96}N^{99}T^{100}R^{101}S^{102}A$ | Region II | Forward: GATTTCAGGCAGAAGTTGCACCCAATGCTGCAGCTGC<br>Reverse: GCAGCTGCAGCATTGGGTGCAACTTCTGCCTGAAATC |
| $N^{122}A$ | Region III | Forward: CATTTTCTGATGTTATTGCTATCAATATTTATAATCC<br>Reverse: GGATTATAAATATTGATAGCAATAACATCAGAAAATG |
| $Y^{126}N^{127}A$ | Region III | Forward: GTTATTAACATCAATATTGCTGCACCATGTGTTGTACCAAATA<br>Reverse: TATTTGGTACAACACATGGTGCAGCAATATTGATGTTAATAAC |
| $N^{122}Y^{126}N^{127}A$ | Region III | Forward: CATTTTCTGATGTTATTGCTATCAATATTGCTGCACC<br>Reverse: GGTGCAGCAATATTGATAGCAATAACATCAGAAAATG |

## Transient complementation assays

Confluent BSC40 cells in 6-well plates were infected with viA28 at a multiplicity of infection (MOI) of 5 PFU per cell and incubated at 37˚C for 1 hour. After replacing the viral inoculum with complete growth media without IPTG, the cells were transfected with 0.05 μg of either vector alone, wild-type A28L, or A28L mutant plasmids using 10 μl of lipofectamine 2000, and then maintained at 37˚C for an additional 24 hours. For mutants that showed diminished A28 expression, $D^{56}R^{58}D^{114}D^{119}A$, $D^{68}K^{72}R^{74}A$ and $G^{90}A$, the transfection condition was modified as the following: we reduced the transfected wild-type A28 expression plasmid DNA from 0.05 μg to 0.01 μg and increased the A28 mutant expression plasmid DNA from 0.05 μg to 2 μg per well to achieve comparable A28 protein expression level between the wild type and the mutant A28 protein of interest. At 24 hpi, the cells were harvested for immunoblot analyses as well as virus titer determination by means of plaque assays in the presence of 100 μM IPTG. These infection-transfection experiments with each A28 mutant plasmid were repeated three times.

## Coimmunoprecipitation of VACV A28 with other EFC components

Freshly confluent BSC40 cells in 100 mm dishes were infected with viA28 at a MOI of 5 PFU per cell for 1 hour at 37˚C, the virus was washed off, and then the cells were immediately transfected with 0.25 μg of either vector, wild-type, or mutant A28L plasmids using 30 μl of lipofectamine 2000 as per the manufacturer's instructions. At 24 hpi, the cells were harvested, washed, and lysed on ice in 500 μl of lysis buffer (0.5% NP-40, 20 mM NaCl, 200 mM Tris pH 8.0, and a protease inhibitor cocktail comprising 2 μg/ml aprotinin, 1 μg/ml leupeptin, 0.7 μg/ml pepstatin, and 1 mM phenylmethylsulfonyl fluoride). The lysates were clarified by centrifugation for 30 minutes at 13,000 rpm and 4˚C, and the supernatants were incubated with 20 μl of 50% Streptavidin slurry (Invitrogen) for 4 hours at 4˚C on a rotating mixer. The mixture was centrifuged, washed four times with lysis buffer, and the bound proteins were eluted with 2.5 mM biotin in PBS plus 0.02% NP-40 after 16–18 hours of incubation at 4˚C. The eluted samples were precipitated with a final concentration of 25% tricarboxylic acid (TCA) on ice for 20 minutes, centrifuged at 14,000 rpm for 10 minutes at 4˚C and washed twice with cold acetone. The protein samples then were heated at 95˚C for 5 minutes, solubilized in SDS-containing sample buffer, and boiled at 95˚C for 5 minutes prior to SDS-PAGE analyses.

## Immunoblot analyses

Whole cell lysate from transient complementation assays and protein samples after coimmunoprecipitations as described above were resolved in SDS-containing sample buffer, separated on SDS-PAGE gels, and transferred onto nitrocellulose membranes for immunoblot analyses using the following antibodies: anti-A16; anti-G9; anti-L1; anti-G3; anti-L5; anti-O3; anti-F9; anti-H3; and anti-D8 antibodies, which have all been described previously [70]. Anti-A28, anti-H2, anti-J5 and anti-A21 rabbit sera were prepared by immunizing rabbits with soluble recombinant proteins, respectively.

## Vaccinia MV-triggered cell fusion-from-without under acidic pH

Cell-cell fusion assays triggered by VACV MV were performed as described previously [70] with slight modification. In brief, GFP- or RFP-expressing HeLa cells were mixed at a 1:1 ratio, seeded in 96-well plates, and cultured overnight. The next day, these cells were pre-treated with 40 μg/ml cordycepin for 1 hour at 37˚C and subsequently infected with the inf/tnf cell lysates harvested from transient complementation assays. After 1 hour of incubation, the

viral inoculum was removed and the cells were washed with warm PBS and then incubated in PBS (pH7.0) or PBS (pH5.0) at 37°C for 3 minutes. Complete growth medium was added to the cells to neutralize the acidic buffer and then replaced with fresh growth medium containing 40 μg/ml cordycepin. These cells were further incubated for 2 hours and subsequently photographed with a MD Image Xpress Micro XL system with a 20x objective. One field in each well was chosen randomly and imaged (596.7 x 596.7 μm) and fluorescence intensity was quantified. The percentage of cell fusion in each well was determined using Fiji software according to the equation: (Surface area of GFP$^+$RFP$^+$ double-fluorescent cells divided by the surface area of single-fluorescent cells) x100%. The experiments were independently repeated three times.

## Statistical analysis

Statistical analyses were performed using Student's t-test in GraphPad Prism v.9.4.1. Data were expressed as mean ±SD. For comparisons with wild-type control, $p$ values were adjusted using the "fdr" method via the "p.adjust" function in R v.4.2.2. Adjusted $p$ values were considered statistically significant at $^*p<0.05$; $^{**}p<0.01$; $^{***}p<0.001$.

## Supporting information

**S1 Fig. 2D NMR HSQC spectrum of VACV fusion protein sA28.** 2D $^1$H/$^{15}$N HSQC spectrum of $^{15}$N-isotope-labeled sA28 (0.2 mM) at pH 6.5. The spectrum was collected at 25°C in an aqueous solution of 20 mM MES containing 50 mM NaCl. The assigned residues are indicated using single-letter codes. The insets show assignment details in the region with the maximum overlap of peaks (red box) and the region for the tryptophan side chain (black box). Assignments of the side-chain NH$_2$ groups from the Asn and Gln residues are indicated with an asterisk, and horizontal gray lines connect the pairs of protons.
(TIF)

**S2 Fig. Normal MV morphogenesis was observed for each VACV A28 mutant under electron microscopy.** Confluent BSC40 cells in a 12-well plate were infected with viA28 and transfected with either control plasmid (No IPTG) or a plasmid expressing either wild-type or mutant A28 plasmids. At 24 hpi, cells were fixed, stained with uranylacetate, dehydrated and epon-embedded for sectioning and EM observation, as described previously [37]. The micrographs were photographed using a Tecnai G2 Spirit TWIN transmission electron microscope operating at 80 kV. All these infected cells produced abundant mature virus particles in cytoplasm with no blockage in virion morphogenesis.
(TIF)

**S3 Fig. Coomassie staining of purified sA28 mutant proteins in SDS-PAGE gels.** Coomassie blue staining of all the recombinant sA28 mutant proteins after purification (for ITC analyses in Fig 9).
(TIF)

**S4 Fig. sA28 mutant proteins are homogenous.** (**A**). Analyses of purified WT and mutant sA28 proteins using gel filtration chromatography. Gel-filtration chromatography profiles for all sA28 mutants were obtained with a TSKgel GMPWXL, 13 μm, all 7.8 mm ID x 30 cm x 4 column (Tosoh Bioscience, PA, USA) and a mobile phase composed of 20mM MES buffer (pH 6.5). All analysis were done at room temperature with a flow rate of 1.0 mL/min and UV detection at 280 nm. (**B**) Analyses of purified WT and mutant sA28 proteins using circular dichroism (CD) spectra analyses described in Materials and Methods. In brief, CD analysis was performed to monitor any aggregation or drastic changes in the secondary structure of

sA28 mutant proteins when compared with the WT sA28. The CD spectrum of sA28 mutants (50 μM) in 20mM MES with 50mM NaCl at pH6.5 were recorded over the wavelength range from 250 to 190 nm, in 0.1 nm steps, as an average of 10 accumulations, using a 1 mm pathlength quartz cuvette on a Jasco-815 spectrometer (Jasco Inc., Japan). All CD analyses were done at 25°C.
(TIF)

**S5 Fig. Comparison of sA28 NMR structure with sA28 in complex with sH2 determined by MD simulation.** For computation analyses, we compared the sA28 NMR structure (Fig 1F) and the sA28 structure (Pink region in Fig 11A) when in a complex with sH2 determined by MD simulation. We observed that partial of α2 helix, β3 and β5 strands became random coils. Therefore, MD simulation suggested that A28 undergoes a mild conformational change when interacting with H2 protein.
(TIF)

**S6 Fig. The MD simulations of wild-type sH2 in the presence of sA28 mutants.** As described in the Materials and Methods, for the sA28-sH2 interaction, one molecule of sA28 and one molecule of sH2 were packed into a cubic box with dimensions ~300 Å using the software PACKMOL [63] to establish the initial point for the MD simulations. Following the same protocol, we conducted a MD simulation of wild-type sH2 in the presence of sA28 mutants, respectively. In contrast to the wild-type sA28 that forms a subcomplex with sH2, the MD simulation data revealed that the four sA28 mutants, i.e., three RI mutants: $D^{61}V^{62}N^{63}D^{64}A$, $D^{68}K^{72}R^{74}A$ and $K^{72}A$, and one RIII mutant: $N^{122}Y^{126}N^{127}A$, could not form a complex with sH2, indicating that these RI and RIII residues are indeed critical for the sA28/sH2 complex formation.
(TIF)

**S7 Fig. VACV sA28 and sH2 structures predicted by AlphaFold.**
(TIF)

**S8 Fig. A role of D119 in RIII of VACV sA28 protein.** In the NMR structure of sA28 protein, D119 forms a close contact with F115 and N122 that in turn contact closely with N124 to help stabilize the short β-strand ranging from N124 to Y126 in Region III (122–127).
(TIF)

**S9 Fig. Alignment of A28L ORF and protein sequences of VACV WR and IA27L virus.** Alignment of A28L DNA and amino acid sequences of VACV WR and IA27L virus showed that the A28 protein encoded in IA27L virus contains 10 extra amino acids, SLAAGRRIQT, at the C-terminus that are absent in the WR virus.
(TIF)

**S10 Fig. Uncropped original images of Fig 5B.**
(TIF)

**S11 Fig. Uncropped original images of Fig 5C.**
(TIF)

**S12 Fig. Uncropped original images of Fig 7B.**
(TIF)

**S13 Fig. Uncropped original images of Fig 7C.**
(TIF)

**S14 Fig. Uncropped original images of Fig 8B.**
(TIF)

**S15 Fig. Uncropped original images of Fig 13B.**
(TIF)

## Acknowledgments

We thank Bernard Moss for providing the IPTG-inducible virus vA28-HAi, Sue-Ping Lee and Wen-Li Pon at the IMB Imaging Core, Academia Sinica for help in imaging data analyses, Hsin-Nan Lin at the IMB Bioinformatics Core, Academia Sinica for assistance in bioinformatic analyses. We also thank Yuan-Chao Lou at the Biomedical Translation Research Center, Academia Sinica for helping with the NMR experiments and data interpretation.

## Author Contributions

**Conceptualization:** Der-Lii Tzou, Wen Chang.

**Data curation:** Der-Lii Tzou.

**Formal analysis:** Chi-Fei Kao, Kathleen Joyce Carillo, Der-Lii Tzou.

**Funding acquisition:** Der-Lii Tzou, Wen Chang.

**Investigation:** Chi-Fei Kao, Min-Hsin Tsai, Kathleen Joyce Carillo.

**Methodology:** Chi-Fei Kao, Min-Hsin Tsai, Kathleen Joyce Carillo, Der-Lii Tzou, Wen Chang.

**Project administration:** Der-Lii Tzou, Wen Chang.

**Resources:** Wen Chang.

**Software:** Kathleen Joyce Carillo.

**Supervision:** Der-Lii Tzou, Wen Chang.

**Validation:** Der-Lii Tzou, Wen Chang.

**Visualization:** Chi-Fei Kao, Kathleen Joyce Carillo.

**Writing – original draft:** Chi-Fei Kao, Kathleen Joyce Carillo, Der-Lii Tzou, Wen Chang.

**Writing – review & editing:** Der-Lii Tzou, Wen Chang.

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
