## [Decision Letter · Decision Letter 0]

28 Aug 2023

Dear Dr. Chang,

Thank you very much for submitting your manuscript "Structural and functional analysis of vaccinia viral fusion complex component protein A28 through NMR and molecular dynamic simulations" for consideration at PLOS Pathogens. As with all papers reviewed by the journal, your manuscript was reviewed by members of the editorial board and by several independent reviewers. The reviewers appreciated the attention to an important topic. Based on the reviews, we are likely to accept this manuscript for publication, providing that you modify the manuscript to address ALL recommendations made by the reviewers.

Sincerely,

Ekaterina E. Heldwein

Guest Editor

PLOS Pathogens

Blossom Damania

Section Editor

PLOS Pathogens

Kasturi Haldar

Editor-in-Chief

PLOS Pathogens

orcid.org/0000-0001-5065-158X

Michael Malim

Editor-in-Chief

PLOS Pathogens

orcid.org/0000-0002-7699-2064

Reviewer Comments (if any, and for reference):

Reviewer's Responses to Questions

**Part I - Summary**

Reviewer #1: One of the least understood aspects of poxvirus biology is its entry fusion mechanism. Poxviruses employ an entry fusion complex (EFC) comprised of eleven proteins, but the EFC structure and molecular mechanism are largely unknown. The current manuscript focuses on the structure and function of one of the key EFC proteins, A28. The authors first used NMR to determine the structure of a soluble form of A28. Since A28 forms a subcomplex with another EFC protein called H2, they then employed molecular dynamics and NMR chemical shift analyses to identify three potential H2 interaction sites in A28. Additional mutagenesis and functional studies showed that two of the potential H2 interaction regions in A28 are important for H2 binding, EFC formation, virus-mediated cell fusion, and viral infection. The structure of A28 is new. The study is important to the field. The data are of good quality and supportive of the conclusion. I am enthusiastic for the publication and only have some relatively minor comments for improving the manuscript.

Reviewer #2: The Authors expressed a truncated form of vaccinia A28 EFC protein, analyzed structure and interaction with known partner H2. NMR structural analysis led to the identification of A28 areas possibly mediating interactions with H2 and those areas were subjected to mutagenesis. Mutant proteins were analyzed for their ability to mediate EFC formation and virus infection/membrane fusion. Two A28 regions were important for complex formation and fusion. Lastly, an A28 area not resolved in the NMR structure was also mutagenized and found to be important for EFC formation and infectivity. In general, the study is well-written, the experiments are well laid out and controlled and the conclusions drawn are largely warranted. The coupling of structure and experimental data, as well as simulation and experimental data, add confidence to the interpretation of their results. The work identifies possible secondary structural domains of A28 likely involved with known interacting partners but larger issues, such as how interactions contribute to EFC fusion, are not yet addressed.

Reviewer #3: Poxviruses are a group of enveloped viruses which contain vaccinia virus (VACV), variola virus, and mpox virus, etc. They can utilize a conserved and enigmatic fusion machinery, called entry-fusion complex (EFC), for post-attachment fusion between viral envelop and cellular membrane. Among the unique proteinaceous components in poxviral EFC, six “ectodomain” structures (L1, F9, G3/L5 sub-complex, and A16/G9 sub-complex) from VACV have currently been experimentally determined. The available structural data demonstrate that the poxviral EFC is not similar to the well-studied type I, II, and III viral fusion proteins in terms of both the number of subunit components and their assembly mode.

The manuscript, presented by Kao et al., describes the structural and functional characterization of one poxviral EFC component -- the A28 subunit. They solved the high-resolution structure of truncated VACV A28 protein (amino acids 38-146) by solution NMR spectroscopy. They determined the affinity between sA28 and its binding partner (sH2) by isothermal titration calorimetry (ITC), and found that these two proteins could form a sub-complex at a 1:1 binding mode. Using intensive in-vitro alanine mutagenesis assays, they identified key residues in A28 that were crucial for membrane fusion.

Overall, the paper is well written and has some merits on the very acute topic during the current mpox outbreak in multiple countries. Although A28 has no homologous structure in the Protein Data Bank, the reported single VACV A28 structure shows an overall fold very similar to that predicted by Alphafold. This will to some extent weaken the innovation of the paper. The article also has some experimental and textual weakness that I think should be addressed.

**Part II – Major Issues: Key Experiments Required for Acceptance**

Reviewer #1: None.

Reviewer #2: None noted.

Reviewer #3: 1. Can the authors provide some experimental or computational evidence on whether the A28 protein undergoes a conformational change after binding to H2. If there is indeed a conformational change in A28, the authors need to take it into account when performing molecular dynamic simulations.

2. We know that manually incorporating mutation(s) on a protein may cause its instability or aggregation. In the in-vitro ITC assays (Figure 8A, 8B and 8C), the authors prepared several mutants of the sA28 protein but lacked the solution-behavior characterization of them. Therefore, more data (such as gel-filtration-chromatography profiles and circular-dichroism-spectra figures, etc.) need to be provided to show that these protein samples are homogeneous.

**Part III – Minor Issues: Editorial and Data Presentation Modifications**

Reviewer #1: 1. The rationale for targeting the specific A28 residues for mutagenesis is not explained clearly. There are many residues in region 1, 2,3. Why did you choose the specific subset of residues for mutagenesis? For example, why do you generate the D56R58D114D119A mutation, which also targets the a2 helix?

2. Line 187: “However, after careful modifications of plasmid transfection conditions”. The specific modification is not described anywhere in the text.

3. Around line 255. The H2 structure used for molecular dynamics analysis is an alphafold predicted structure. It should be stated as such in the result section.

4. Line 550-556. The method for fusion from without experiment appears to be incorrect. Why were the cells incubated with inf/tf lysates overnight?

5. Line 640. There is an extra B in the text.

6. Figure 1D. There is a beta5 strand in the figure, but the text only describes 4 beta strands.

7. Figure 2D. Region 2 does not have signal decay and was later shown not important for H2 binding. Perhaps the data in Figure 2D is a better predictor for H2 binding region?

8. Figure 6. For N99T100R101S102A mutation, cell fusion and co-IP experiments are not performed.

9. Figure 7. For N122A and Y126N127A mutations, cell fusion and co-IP experiments are not performed.

Reviewer #2: 1. Is A28 thought to be mainly a structural protein and involved in complex formation? There do not appear to be A28 mutations that affect fusion without affecting complex formation (thus far). Presumably, one or more of the EFC proteins will contain domains involved in membrane fusion but perhaps not directly in complex formation. Mutations with the aforementioned attributes would lead to a better idea about membrane fusion outside of pure complex formation. This idea could be discussed.

2. Were H2/A28 MD simulations repeated with any A28 mutants where complex formation was disrupted? What would be the expected result?

3. Alpha fold structural predictions are mentioned in the discussion. It would be of interest to show those for A28 and H2, especially to provide support for certain domains that seem to have structural functions. Could be placed in the supplement if high quality predictions can be obtained.

4. Methods: More information on how the virus fusion-from-without assay quantitation was conducted would be helpful. For instance, was the entire well imaged or random fields? How many fields?

5. “(B).” at end of Figure 4 legend. Not needed?

Reviewer #3: 1. The protein nomenclatures of VACV are not identical to mpox counterparts. E.g., A28 in VACV is A30 in mpox virus. Therefore, it may be necessary to modify the protein names of mpox virus involved in the paper or provide additional explanations in introduction section.

2. The secondary structure of A28 is inconsistent in the text and the figures: the α3-helix in Figure 1C is forgotten to label, and the β5-strand is present in Figure 1C and 1D but is not described or missing in the text (lines 96-99 and lines 581-584) and Figure 4.

3. I noticed that the SDS-PAGE-gel and immunoblot images in the manuscript were cropped to a single lane with small size, which makes almost all the images in figure panels spliced. Therefore, the authors may need to include the uncropped original images of these figures in supplementary materials.

4. The scale bars should be added to these cell-cell fusion images: Figure 5B, Figure 6B, Figure 7B, and Figure 10B.

5. In Figure 8A and 8B, the vertical axes of some differential power curves lack scale unit values.

6. The authors discussed that some anti-A28 antibodies may interfere with H2 binding to A28 through steric hindrance (lines 385-390). While previously studies have shown that the neutralizing antibody response could be enhanced when A28 and H2 are used together as a complex (Shinoda, K. et al., Virology, 2010, 405, 41-49). Therefore, more discussion should be included in the manuscript.

PLOS authors have the option to publish the peer review history of their article (what does this mean?). If published, this will include your full peer review and any attached files.

Reviewer #1: No

Reviewer #2: No

Reviewer #3: No

Figure Files:

Data Requirements:

Reproducibility:

References:

---

## [Editor Report · Decision Letter 1]

31 Oct 2023

Dear Dr. Chang,

We are pleased to inform you that your manuscript 'Structural and functional analysis of vaccinia viral fusion complex component protein A28 through NMR and molecular dynamic simulations' has been provisionally accepted for publication in PLOS Pathogens.

Best regards,

Ekaterina E. Heldwein

Guest Editor

PLOS Pathogens

Blossom Damania

Section Editor

PLOS Pathogens

Kasturi Haldar

Editor-in-Chief

PLOS Pathogens

orcid.org/0000-0001-5065-158X

Michael Malim

Editor-in-Chief

PLOS Pathogens

orcid.org/0000-0002-7699-2064
---

## [Editor Report · Acceptance letter]

6 Nov 2023

Dear Dr. Chang,

We are delighted to inform you that your manuscript, "Structural and functional analysis of vaccinia viral fusion complex component protein A28 through NMR and molecular dynamic simulations," has been formally accepted for publication in PLOS Pathogens.

Best regards,

Kasturi Haldar

Editor-in-Chief

PLOS Pathogens

orcid.org/0000-0001-5065-158X

Michael Malim

Editor-in-Chief

PLOS Pathogens

orcid.org/0000-0002-7699-2064